# Stabilized $Fe_7C_3$ catalyst with K–Mg dual promotion for robust $CO_2$ hydrogenation to high-value olefins

Fei Qian[1,2,3,6], Maolin Wang [4,6], Zidu Wei[2], Yi Cai[1,2,3], Zeping Sun[2], Ruikang Liang[2], Guangbo Liu[5], Ming Qing[2], Hong Wang[2], Jinjia Liu[1,2] ✉, Xing-Wu Liu [1,2] ✉, Yong Yang[1,2,3] & Xiao-Dong Wen [1,2,3] ✉

Iron carbide catalysts, particularly the $Fe_7C_3$ phase, hold significant potential for efficient $CO_2$ hydrogenation to olefins, yet stabilizing this phase under reactive conditions remains a major challenge. Herein, we report a robust and efficient synthesis of nearly phase-pure $Fe_7C_3$ catalysts derived from Prussian blue analogues, whose stability is significantly enhanced by strategically incorporating K and Mg promoters. Comprehensive characterization reveals that K accelerates the carbonization process and markedly enhances olefin selectivity, whereas Mg effectively suppresses water-induced oxidation, preserving the structural integrity of the $Fe_7C_3$ phase. Under optimized reaction conditions (340 °C, 2 MPa, $H_2/CO_2 = 3$), the $Fe_7C_3$-KMg catalyst achieves a high $CO_2$ conversion of 41.5% and an olefin selectivity of 67.1%, maintaining exceptional catalytic stability for over 1000 hours. These findings offer valuable new insights into the rational design of robust iron carbide catalysts for sustainable and efficient $CO_2$ conversion into high-value chemicals.

Hydrogenation of $CO_2$ into hydrocarbons has attracted intense interest as a route to produce value-added fuels[1,2] and chemicals[3,4] while mitigating $CO_2$ emissions[5–9]. Modified iron-based Fischer–Tropsch synthesis (FTS) catalysts are particularly appealing for this reaction because of their activity, affordability, and inherent promotion of the water–gas shift reaction (WGS)[10–12]. Under typical $CO_2$ hydrogenation conditions (300 – 360 °C, 1 – 5 MPa), iron catalysts tend to evolve into a mixture of iron carbide and iron oxide phases[13]. In fact, it is commonly observed in operando studies that iron initially carburizes to form carbides (e.g. $\chi$-$Fe_5C_2$)[11,14], but the $H_2O$ byproduct of $CO_2$ conversion concurrently re-oxidizes some iron into magnetite ($Fe_3O_4$)[13]. Thus, the catalyst reaches a dynamic steady state comprising iron carbide and oxide phases, which synergistically couple FTS and reverse WGS. Consistent with this understanding, the vast majority of

literature reports on Fe-based $CO_2$ hydrogenation catalysts have identified Hägg iron carbide ($\chi$-$Fe_5C_2$) as the predominant active phase, with magnetite ($Fe_3O_4$) playing a secondary but important role in maintaining activity[11,14].

Emerging evidence underscores the catalytic importance of $Fe_7C_3$. Zhao et al. showed that all major iron carbides, including $Fe_7C_3$ are catalytically active in FTS[15,16], and Chang et al. reported that $Fe_7C_3$ can exhibit the highest intrinsic turnover frequency among the iron carbides under typical FTS conditions[17]. These findings suggest that $Fe_7C_3$ could offer performance advantages over the traditional iron carbide phases, surpassing even $\chi$-$Fe_5C_2$ in intrinsic activity, and thus merits special attention in catalyst design. In contrast, the role of the $Fe_7C_3$ phase in $CO_2$ hydrogenation has remained largely unexplored[18], primarily due to a historical research

[1]State Key Laboratory of Coal Conversion, Institute of Coal Chemistry, Chinese Academy of Sciences, Taiyuan 030001, China. [2]National Energy Center for Coal to Liquids, Synfuels China Co. Ltd., Huairou District Beijing 101400, China. [3]University of Chinese Academy of Sciences, No. 19A Yuquan Road, Beijing 100049, PR China. [4]Beijing National Laboratory for Molecular Sciences, New Cornerstone Science Laboratory, College of Chemistry and Molecular Engineering, Peking University, Beijing, China. [5]Key Laboratory of Photoelectric Conversion and Utilization of Solar Energy, Qingdao Institute of Bioenergy and Bioprocess Technology, Chinese Academy of Sciences, Qingdao 266101, China. [6]These authors contributed equally: Fei Qian, Maolin Wang. ✉e-mail: liujinjia@synfuelschina.com.cn; liuxingwu@sxicc.ac.cn; wxd@sxicc.ac.cn

focus on $\chi$-$Fe_5C_2$ and $Fe_3O_4$, as well as the practical challenge of obtaining $Fe_7C_3$ as a stable active phase under reaction conditions. Recent studies have begun to probe $Fe_7C_3$-based catalysts for $CO_2$ hydrogenation, though with mixed[19]. Pasupulety et al. examined a K and Zn promoted iron catalyst supported on $ZrO_2$ synthesized via a citric acid method with the aim of facilitating $Fe_7C_3$ formation. They reported that a $CO/H_2$ pretreatment could indeed produce the $Fe_7C_3$ phase in this Fe–Zn–K/$ZrO_2$ system; however, the activated catalyst was far from phase-pure, as a significant fraction of magnetite remained present alongside $Fe_7C_3$. The persistence of substantial $Fe_3O_4$ as well as the dominant $ZrO_2$ support phase corresponded with suboptimal catalytic performance. The underwhelming results from such attempts highlight the need for new approaches to generate a purer and more stable $Fe_7C_3$ phase under $CO_2$ hydrogenation conditions.

The scarcity of stable $Fe_7C_3$ in working catalysts arises from several intertwined factors. First, the thermodynamic and kinetic constraints governing phase formation often favor more common iron carbides or metallic iron under typical reaction temperatures and $CO/H_2$ ratios[20,21]. Second, the complexity of $Fe_7C_3$'s formation from precursor materials often necessitates stringent pretreatment conditions that are not easily translated to large-scale or continuous operations[17]. Third, even when $Fe_7C_3$ is formed, maintaining its phase purity and stability under highly dynamic reaction environments, such as high partial pressures of water and oxidizing conditions inherent in $CO_2$ hydrogenation, poses additional hurdles[13,22].

In light of the gaps in prior research, we focus on $Fe_7C_3$ as a target active phase for $CO_2$ hydrogenation and seeks to overcome the aforementioned stabilization challenges[21]. In contrast to previous works, we demonstrate that an in-situ carburization strategy can produce a nearly phase-pure $Fe_7C_3$ catalyst that remains stable under reaction conditions for prolonged periods. Notably, the $Fe_7C_3$-rich phase achieved in our work exhibits enhanced catalytic activity, including higher $CO_2$ conversion and $C_{2+}^-$ olefin productivity compared to conventional $Fe_3O_4$/$\chi$-$Fe_5C_2$-based catalysts. These findings represent the first clear evidence that $Fe_7C_3$ can serve as a durable and highly active phase for $CO_2$ hydrogenation to hydrocarbons. Accordingly, the motivation of this study is to fill the knowledge gap regarding $Fe_7C_3$ by exploring its formation and function during $CO_2$ hydrogenation. In this work, we aim to highlight a new pathway for designing iron-based $CO_2$ hydrogenation catalysts beyond the traditional $Fe_3O_4$/$\chi$-$Fe_5C_2$ paradigm, thereby advancing the development of more efficient $CO_2$-to-fuels technologies.

## Results
### Structural characterizations
Following our previously reported method for synthesizing Prussian blue analogue (PBA) precursors[23], a series of PBA-based catalysts were synthesized and activated under an $NH_3$ atmosphere. This $NH_3$ atmosphere not only promotes the thermal decomposition of the PB analogues but also facilitates the formation of $Fe_2N$ as the primary phase in the fresh catalysts, as confirmed by X-ray diffraction (XRD) (Supplementary Fig. 1 and Fig. 2). Once subjected to $CO_2$ hydrogenation, $Fe_2N$ rapidly undergoes an in situ transformation into iron oxide and iron carbide[24], as illustrated in Fig. 1a. Specifically, the Fe and FeMg catalysts both display characteristic $Fe_3O_4$ peaks, indicating extensive oxidation of iron under the reaction conditions. Introducing potassium (FeK) induces partial carbide formation, and the subsequent addition of magnesium (FeKMg) leads to the clear emergence of a distinct $Fe_7C_3$ phase. Mössbauer spectroscopy and fitting analyses (Fig. 1b, Supplementary Fig. 3, Supplementary Table 1) corroborate the presence of $Fe_7C_3$ and reveal the composition of each spent catalyst[25].

To gain deeper insights into the electronic structure, we employed X-ray absorption spectroscopy (XAS) and X-ray photoelectron spectroscopy (XPS). Fe K-edge XANES spectra (Fig. 1c) reveal that the spent Fe and FeMg catalysts have a pre-edge feature indicative of iron oxides. Introducing K shifts the absorption edge between oxidized and metallic states, while subsequent Mg addition drives a more pronounced shift toward lower energy, closely matching that of $Fe_7C_3$. This suggests that, under reaction conditions, Fe predominantly exists as the iron carbide in FeKMg[11,26]. The XAFS fitting results show that Fe and FeMg catalysts have the similar short-range structure with a Fe-O scattering path at 3.5 Å, whereas the presence of FeKMg maintains a resembling coordination number structure closer to $Fe_7C_3$ (Fig. 1d, Supplementary Fig. 4, Supplementary Table 2). Consistent with these findings, XPS results (Fig. 1g) show that Fe and FeMg retain oxidized surfaces, whereas FeK exhibits partial metallic character and FeKMg is dominated by metallic Fe signals[27]. Catalyst compositions (Supplementary Table 3) indicate that all catalysts possess similar Fe contents.

High-resolution transmission electron microscopy (HRTEM) shows the average crystallite size of the FeKMg catalyst is about 71.2 nm (Fig. 1e) and a lattice spacing of ~2.1 Å corresponding to the (102) plane of $Fe_7C_3$ (Fig. 1f). STEM-EDS mapping and line (Fig. 1h–i) scans indicate uniform element distribution. For comparison, TEM images of the spent Fe catalyst (Supplementary Fig. 5) reveal a uniform distribution of Fe and O (average particle size ~47.06 nm). Similarly, STEM-EDS mapping of FeK (Supplementary Fig. 6) and FeMg (Supplementary Fig. 7) show homogeneous elemental distributions. However, FeMg remains mostly $Fe_3O_4$, whereas FeK is a mixture of $Fe_7C_3$ and $Fe_3O_4$.

Figure 2 illustrates the structural evolution of the catalysts during $CO_2$ hydrogenation. The initial $Fe_2N$ phase in Fe and FeMg catalysts (Fig. 2a and b) rapidly oxidizes to iron oxides within 8 hours. In contrast, FeK (Fig. 2c) initially undergoes carbonization to form iron carbides, but subsequently experiences partial re-oxidation, resulting in a mixed oxide–carbide phase. Notably, only FeKMg (Fig. 2d) maintains a stable carbide structure throughout the reaction, predominantly featuring the rarely reported $Fe_7C_3$ phase. Mössbauer quantification (Fig. 2e, Supplementary Fig. 8, Supplementary Table 4) confirms that potassium significantly promotes the carbonization process, while magnesium plays a key role in stabilizing the $Fe_7C_3$ phase. Figure 2f schematically summarizes the distinct roles of K and Mg during phase evolution. In our system, a small amount of $Fe_7C_3$ pre-exists in the fresh catalyst and acts as a nucleation base for further carburization. The metastable $Fe_2N$ gradually converts into $Fe_7C_3$ under the assistance of potassium. Meanwhile, magnesium plays a crucial role in maintaining the structural and chemical stability of the catalyst during $CO_2$ hydrogenation. The thermodynamic phase diagram for iron carbide formation was simulated through theoretical calculations. Under low carbon chemical potential, the formation energies of different iron carbides are very close, suggesting a flat potential energy surface that facilitates facile phase transformations among them. The presence of $Fe_7C_3$ under these conditions indicates its thermodynamic viability and the possibility of stabilization even in carbon-lean environments (Supplementary Fig. 9).

Previous catalyst systems predominantly rely on the coexistence and dynamic equilibrium between $\chi$-$Fe_5C_2$ and $Fe_3O_4$ phases, identifying $\chi$-$Fe_5C_2$ as the primary active component[9]. In clear contrast, our FeKMg catalyst achieves nearly phase-pure $Fe_7C_3$ stabilization under realistic $CO_2$ hydrogenation conditions. Comprehensive characterization methods explicitly confirm $Fe_7C_3$ as the dominant and stable catalytic phase, providing deeper mechanistic insights into the active site characteristics of iron carbide catalysts. Additionally, surface-sensitive XPS analysis detects minor dispersed $Fe^{3+}$ oxide species (Fig. 1g) that remain undetectable in bulk analyses, suggesting these trace oxide species could beneficially contribute RWGS activity by forming highly dispersed and advantageous $Fe_xO$ sites on $Fe_7C_3$ surface.

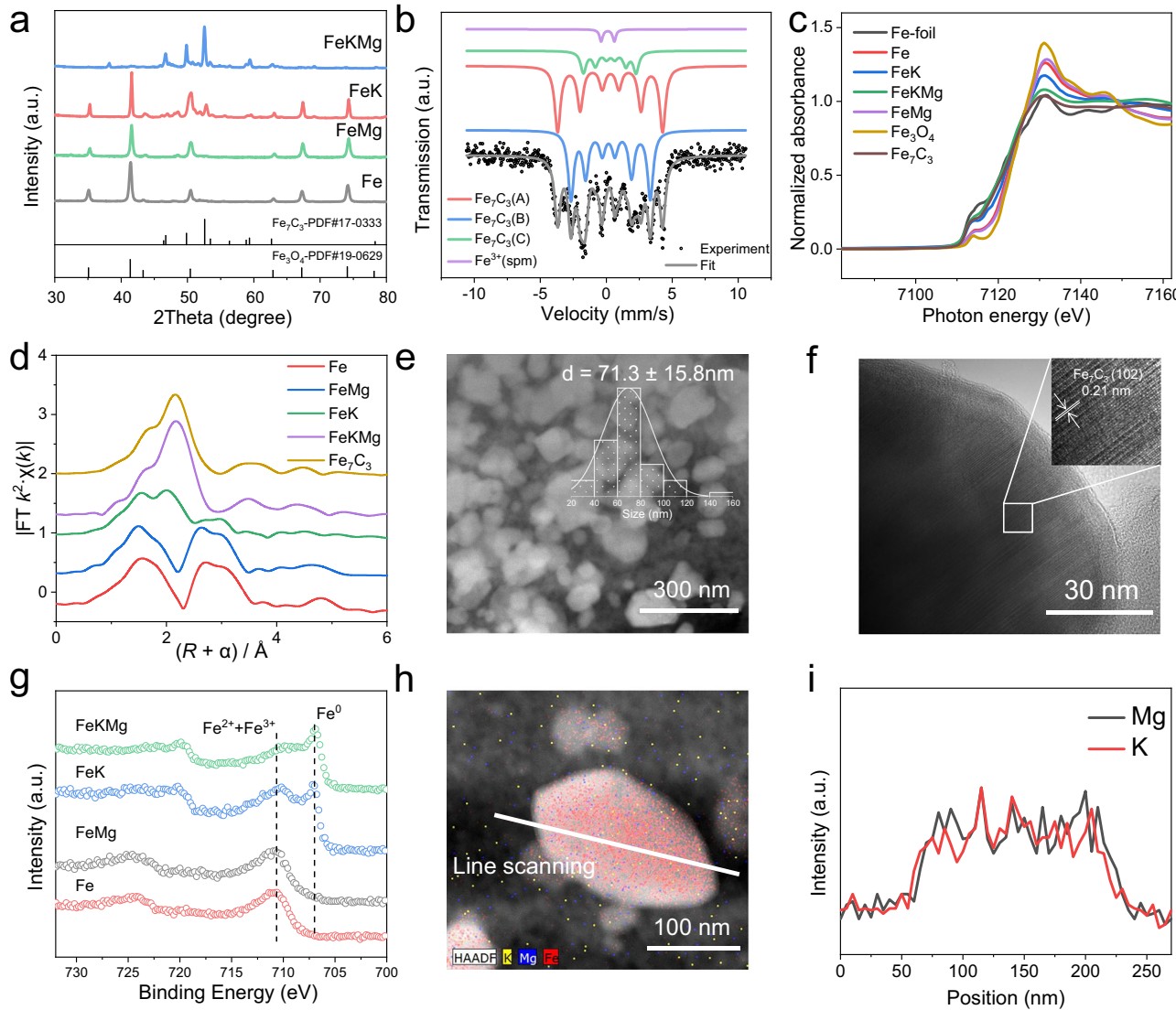

**Fig. 1 | Comprehensive characterization of spent catalysts.** XRD patterns of the spent Fe, FeMg, FeK, and FeKMg (**a**). Mössbauer spectrum of the spent FeKMg (**b**). Fe K-edge X-ray absorption near-edge structure (XANES) spectra (**c**). Fourier transform extended X-ray absorption fine structure (FT EXAFS) at the Fe K-edge (**d**) for the Fe, FeMg, FeK, and FeKMg. Particle size distribution (**e**) and HRTEM image (**f**) of the FeKMg catalyst. Fe 2*p* XPS profiles for the Fe, FeMg, FeK, and FeKMg catalysts (**g**). STEM-EDS elemental mapping (**h**) and line scanning analysis (**i**) of the FeKMg catalyst. All spent catalysts (Fe, FeMg, FeK, FeKMg) were collected after reaching steady-state catalytic performance under standard conditions (340 °C, 2 MPa, $H_2/CO_2 = 3$, GHSV = 6 L·$g_{cat}^{-1}$·$h^{-1}$).

## Catalytic performance

Beyond their structural differences, these catalysts exhibited notable disparities in catalytic behavior. To clearly demonstrate these differences, we compared their catalytic performance over the first 24 hours of time on stream (TOS) under identical reaction conditions. Table 1 summarizes the corresponding results, while Fig. 3a,b and Supplementary Fig. 10 showed the associated product distributions. The Fe and FeMg catalysts displayed similar behavior, each yielding a high fraction (~ 50%) of low-value products ($CH_4$ and CO). Upon the introduction of potassium (FeK), olefin selectivity improved markedly, with high-value olefins reaching approximately 49.2%. Notably, despite initially high $CO_2$ conversion across all four catalysts (Supplementary Fig. 11), catalysts without Mg gradually deactivated, concomitant with increased CO formation. This observation, together with structural analyses, indicates that the depletion of iron carbides, particularly $Fe_7C_3$, is a critical factor behind the decreasing FTS activity in later stages. In contrast, FeKMg consistently delivered higher $CO_2$ conversion and lower selectivity to $C_1$ by-products, ultimately achieving around 67.1% selectivity to high-value olefins.

We next systematically explored the influence of reaction conditions and Mg loading on catalytic performance. Varying the Mg/Fe ratio led to an initial increase in $CO_2$ conversion, followed by a decrease (Fig. 3c, Supplementary Table 5). The optimum Mg/Fe ratio of 0.04 yielded both the highest conversion and relatively low $C_1$ by-product selectivity. XRD analyses (Supplementary Figs. 12 and 13, Supplementary Table 6) showed no significant change in catalyst phase with different Mg contents. $Fe_7C_3$ persisted as the main active phase, although $MgCO_3$ appeared at higher Mg loadings. An elevated reaction temperature enhanced $CO_2$ conversion but also boosted $C_1$ by-product formation, while increasing the space velocity diminished conversion and concurrently raised the $C_1$ fraction (Supplementary Fig. 14, Supplementary Tables 7 and 8). The $Fe_7C_3$-KMg catalyst exhibited stable performance across varying $H_2/CO_2$ ratios, and post-reaction XRD confirmed that $Fe_7C_3$ remained the dominant phase without notable $Fe_3O_4$ formation (Supplementary Fig. 15). Compared with previously reported catalysts (Fig. 3d, Supplementary Table 9), our $Fe_7C_3$-KMg catalyst demonstrates significantly higher olefin selectivity and minimal production of low-value by-products such as $CH_4$ and CO. Under

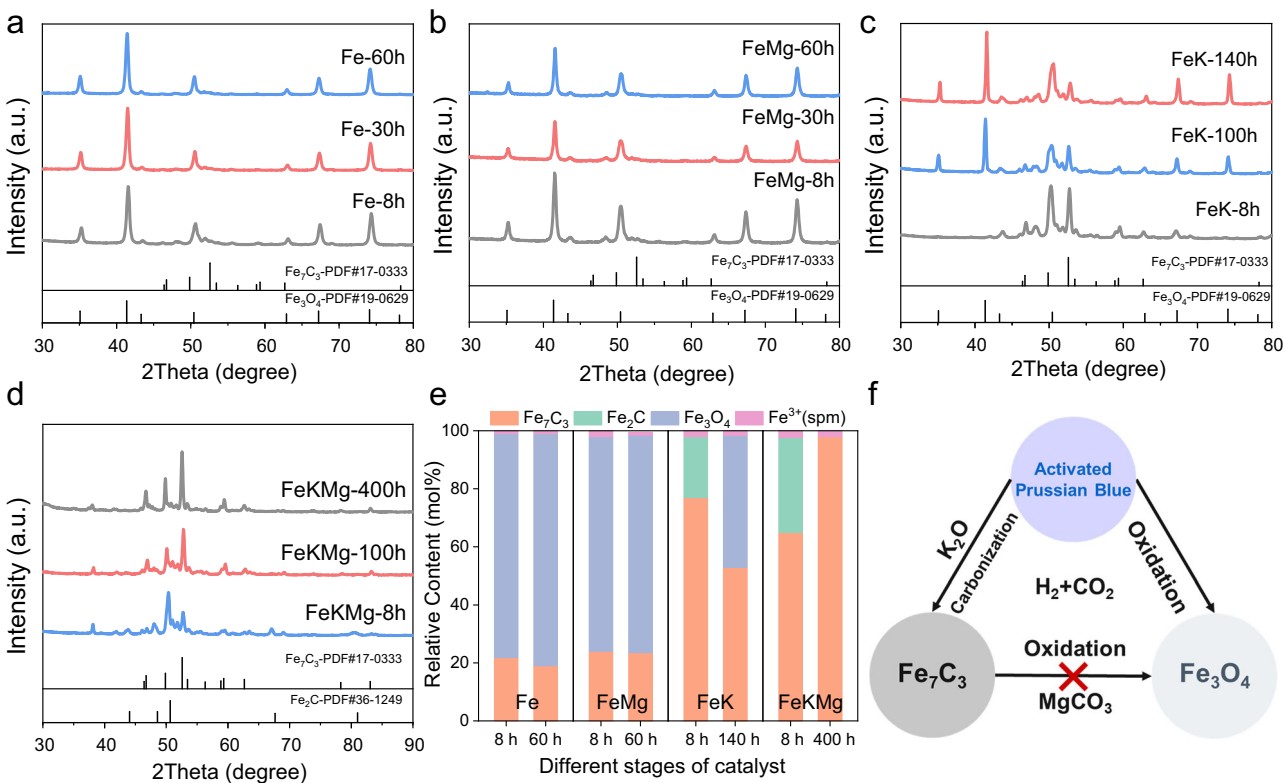

**Fig. 2 | Phase evolution of catalyst during the reaction. a** Fe, **b** FeMg, **c** FeK, **d** FeKMg. Reaction conditions: 0.10 g catalyst, 340 °C, GHSV = 6 L·g$_{cat}^{-1}$·h$^{-1}$, 2 MPa, H$_2$/CO$_2$ = 3. Overview of the catalyst phase transitions observed during the reaction (**e**). Schematic illustration of promoters' effect on catalyst phases (**f**).

extended operation (340 °C, 2.0 MPa, H$_2$/CO$_2$ = 3, GHSV = 6 L·g$_{cat}^{-1}$·h$^{-1}$), Fe$_7$C$_3$-KMg maintained exceptional stability over 1000 hours (Fig. 3e), exhibiting negligible changes in conversion and product distribution. Specifically, relative to recently reported alkali and alkaline-earth metal promoted iron catalysts[12,19], our catalyst achieves comparable CO$_2$ conversion (~41.5%) at notably milder conditions (2 MPa vs. 3 MPa) with superior olefin selectivity (67.1%) and enhanced long-term stability. This outstanding performance primarily results from the high phase purity of Fe$_7$C$_3$, distinctly superior to commonly reported χ-Fe$_5$C$_2$/Fe$_3$O$_4$ mixtures[9], and from the synergistic promoting effects of potassium and magnesium. These characteristics underscore its robust potential for practical CO$_2$ hydrogenation applications.

In summary, our results provide clear evidence that nearly phase-pure Fe$_7$C$_3$ catalysts derived from PBAs can efficiently drive CO$_2$ hydrogenation to olefins with outstanding selectivity and long-term stability. These performance differences among catalysts underscore the critical role of phase purity and promoter interactions. The underlying structural mechanisms driving these catalytic outcomes, especially the stabilization effect of Mg and promotion effect of K, will be discussed in depth in subsequent sections.

## Structural information and mechanism of the magnesium promoter

To elucidate the structure-performance relationship in magnesium-promoted Fe$_7$C$_3$ catalysts, a systematic investigation of the magnesium species' structure and spatial distribution within the catalytic system is critically needed. XPS quantification reveals that magnesium is predominantly localized on the catalyst surface, exhibiting a notably higher Mg/Fe atomic ratio compared to the bulk (Supplementary Fig. 16). Further XPS analyses confirm that Mg and K promoters primarily exist as surface MgCO$_3$ and K$_2$O species, respectively (Supplementary Fig. 17–18). Specifically, the characteristic K 2$p$ peak around 292.7 eV clearly indicates the presence of surface-bound potassium

oxide (K$_2$O) rather than metallic potassium or lattice-incorporated potassium species, aligning well with literature consensus on potassium-promoted iron carbide catalysts[28,29]. Additionally, combined XPS analyses and DFT calculations indicate the absence of significant electronic interactions between the potassium and magnesium promoters (Supplementary Fig. 19).

Considering the critical influence of water formed during CO$_2$ hydrogenation on catalyst stability, we specifically examined the impact of water exposure on catalyst performance (Fig. 4a and Supplementary Fig. 20). Whereas introducing H$_2$O significantly decreased the activity of FeK, the FeKMg catalyst exhibited only minor performance fluctuations, indicating that Mg incorporation critically enhances catalyst resistance to water-induced deactivation. Even under high water partial pressures, FeKMg exhibited exceptional structural and catalytic stability. Raman spectra (Fig. 4b) corroborate this finding: although the FeK catalyst progressively lost surface carbon species and displayed emerging Fe–O vibrational peaks, FeKMg showed no notable oxidation-related signals.

To further elucidate the interactions between H$_2$O and the catalyst, we performed D$_2$O-TPD experiments (Supplementary Fig. 21) and transient kinetic analyses (Supplementary Fig. 22). TPD measurements revealed that both FeK and FeKMg catalysts exhibit similar D$_2$O adsorption capacities; however, FeK produced a noticeably higher signal for dissociated D$_2$. This suggests that the addition of Mg inhibit the dissociation of D$_2$O. Moreover, we carried out transient kinetic experiments to quantify the catalyst's capacity for D$_2$O dissociation. By monitoring the D$_2$ signal generated from the reaction between CO and surface-dissociated D$_2$O, we were able to determine the amount of dissociated D$_2$O on the catalyst surface. The FeKMg catalyst showed a lower quantity of dissociated D$_2$O, further confirming that Mg suppresses D$_2$O dissociation. This reduction in water splitting is a key factor in enhancing the stability of the FeKMg catalyst under reaction conditions, as it protects the active iron carbide phase from oxidation.

**Table 1 | Catalytic performance for CO₂ conversion over catalysts**

| Catalysts | Phase Composition [wt%] | CO₂ Conversion [%] | C₁ sel.[mol%] | | C₂₋₄ sel. [mol%] | C₅⁺ sel. [mol%] | C₂₋₄ O/P | C₂₊⁼ sel. [mol%] |
|---|---|---|---|---|---|---|---|---|
| | | | CO | CH₄ | | | | |
| Fe | 80%Fe₃O₄-19%Fe₇C₃ | 33.7 | 9.1 | 50.6 | 33.6 | 6.7 | 1.4 | 23.9 |
| FeMg | 74%Fe₃O₄-24%Fe₇C₃ | 34.0 | 8.0 | 47.5 | 36.2 | 8.3 | 1.2 | 25.8 |
| FeK | 46%Fe₃O₄-53%Fe₇C₃ | 35.9 | 28.6 | 12.9 | 36.4 | 22.1 | 6.6 | 49.2 |
| FeKMg | 98%Fe₇C₃ | 41.5 | 12.4 | 10.5 | 37.4 | 39.6 | 7.9 | 67.1 |

Reaction conditions: 340 °C, 2 MPa, 0.1 g catalyst, GHSV = 6 L·g$_{cat}^{-1}$·h$^{-1}$, H₂/CO₂ = 3.

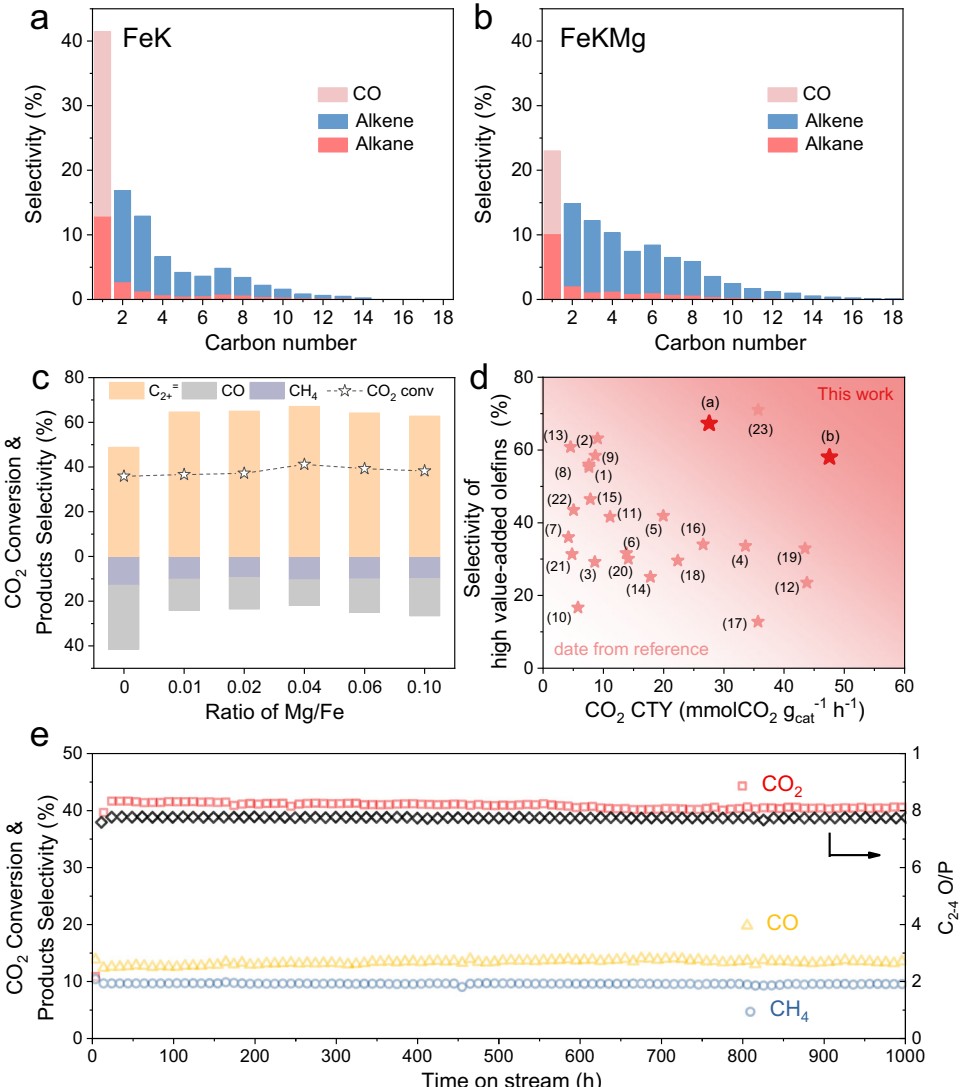

**Fig. 3 | Catalytic performance.** Product distributions from CO₂ hydrogenation over Fe-based catalysts: **a** FeK, and **b** FeKMg. Bar charts show hydrocarbon selectivity versus carbon number. **c** CO₂ conversion and product selectivity for Fe₇C₃-K with varying Mg levels. **d** Comparison of the catalytic performance of Fe₇C₃ with that of other previously reported catalysts (These catalysts cited from 1 to 23 are shown in supplementary Tables 8, high-value olefins: olefins of all carbon numbers). **e** Stability test of the Fe₇C₃-KMg catalyst. (Reaction conditions: 0.10 g of catalyst, 340 °C, 2.0 MPa, H₂/CO₂ = 3, GHSV = 6 L·g$_{cat}^{-1}$·h$^{-1}$).

Both FeK and FeKMg catalysts exhibited remarkably high olefin selectivity. Previous studies have demonstrated that alkali metal promoters, particularly potassium, effectively enhance olefin selectivity by suppressing secondary olefin hydrogenation[30,31]. Our pulse experiment using propylene hydrogenation (Supplementary Fig. 23) explicitly confirmed this effect, clearly showing that potassium significantly inhibits olefin hydrogenation. In contrast, magnesium had minimal influence on this reaction. Thus, our findings indicate that the superior

olefin selectivity observed is primarily attributed to the potassium promoter's ability to effectively limit secondary hydrogenation of olefins.

To clarify the atomic-scale mechanism underlying magnesium's stabilizing effect, we performed DFT calculations to investigate water dissociation behavior on various catalyst surfaces. The results for Fe₇C₃ and Fe₇C₃-Mg are presented in Supplementary Fig. 24, while Fig. 5a shows the calculations for Fe₇C₃-K and Fe₇C₃-KMg. Adsorption

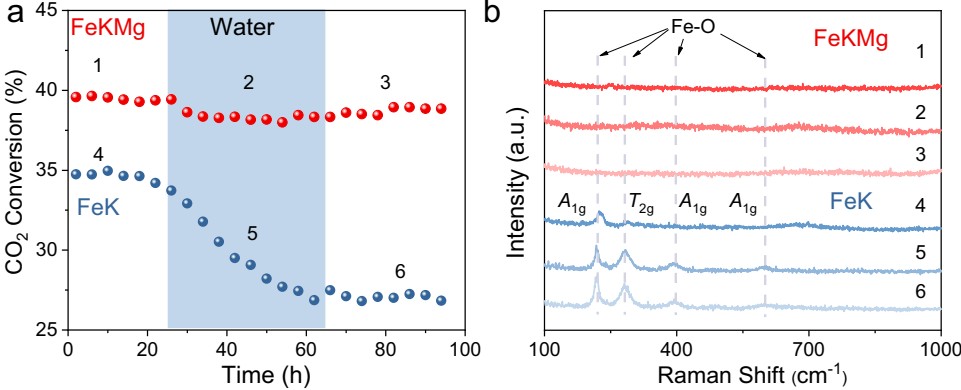

**Fig. 4 | Effect of water on catalytic performance and surface structure. a** $CO_2$ hydrogenation activity under dry conditions and with an additional 10% gas-phase water (Reaction conditions: 0.10 g catalyst, 340 °C, 2.0 MPa, $H_2/CO_2 = 3$, GHSV = 6 $L \cdot g_{cat}^{-1} \cdot h^{-1}$). **b** Corresponding Raman spectra recorded after each reaction stage (spectra 1–6 represent sequential stages of reaction).

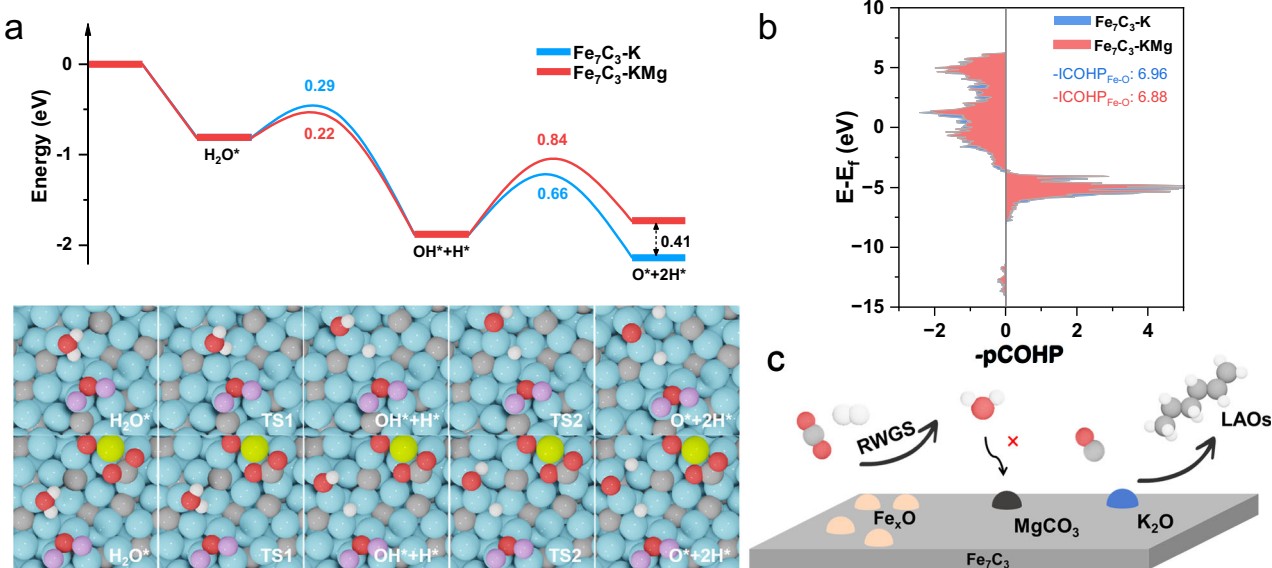

**Fig. 5 | Mechanism of the Mg promoter. a** Theoretical calculations in dissociation process of $H_2O$ on $Fe_7C_3$-K and $Fe_7C_3$-KMg. **b** Crystal orbital Hamilton population (COHP) analysis of Fe-O bonds on $Fe_7C_3$-K and $Fe_7C_3$-KMg. **c** Schematic illustration of the olefins synthesis reaction process via the $Fe_7C_3$-KMg catalyst system from $CO_2$.

energies of water on all four models are comparable, averaging approximately -0.80 eV, indicating similar water-binding strengths. Calculated energy barriers for the first O–H bond dissociation were 0.29 eV on $Fe_7C_3$-K and 0.22 eV on $Fe_7C_3$-KMg, indicating that initial dissociation occurs readily under reaction conditions. However, the subsequent dissociation of the second O–H bond is significantly more challenging, especially on $Fe_7C_3$-KMg, with an energy barrier increasing to 0.84 eV compared to 0.66 eV on $Fe_7C_3$-K. These calculations clearly illustrate magnesium's substantial inhibitory effect on O–H bond cleavage.

Moreover, introducing magnesium significantly hampers oxygen adsorption, as evidenced by calculated reaction energies for water dissociation: −2.14 eV on $Fe_7C_3$-K and −1.73 eV on $Fe_7C_3$-KMg. Crystal orbital Hamilton population (COHP) analysis, which quantitatively measures bond strength through integrated COHP (-ICOHP) values (higher values indicate stronger bonding), further corroborates this observation. As depicted in Fig. 5b, the -ICOHP values for Fe–O bonds decrease from 6.96 on $Fe_7C_3$-K to 6.88 on $Fe_7C_3$-KMg, aligning with the reaction energy trends. Supplementary Fig. 25 shows analogous results for $Fe_7C_3$ and $Fe_7C_3$-Mg, with -ICOHP values of 7.03 and 6.95, respectively. These results collectively suggest that magnesium markedly

reduces the oxygen adsorption capacity on $Fe_7C_3$ surfaces. Charge density difference analyses (Supplementary Fig. 26) further reveal electronic interactions: electrons transfer from $K_2O$ to $Fe_7C_3$ in $Fe_7C_3$-K, creating an electron-rich $Fe_7C_3$ surface favorable for oxygen adsorption and subsequent oxidation. Conversely, $MgCO_3$ in $Fe_7C_3$-Mg acts as an electron acceptor, resulting in a charge-deficient $Fe_7C_3$ surface. Bader charge analysis quantifies this electron depletion, showing $Fe_7C_3$ loses approximately 1.03 electrons upon magnesium incorporation, thereby significantly mitigating oxygen adsorption and subsequent surface oxidation.

Integrating these findings, we propose the following catalytic mechanism (Fig. 5c): $CO_2$ initially undergoes the RWGS reaction, facilitated by highly dispersed $Fe_xO$ species on the $Fe_7C_3$ catalyst surface, producing CO and water. Magnesium critically inhibits dissociation of the generated water, effectively preventing oxidation and subsequent deactivation of the active $Fe_7C_3$ phase. Consequently, in the presence of potassium promotion, the stable $Fe_7C_3$ surface efficiently catalyzes the conversion of CO and $H_2$ into high-value olefins. This synergistic interplay between magnesium stabilization and potassium promotion underlies the outstanding selectivity and long-term stability of the $Fe_7C_3$-KMg catalyst. Nevertheless, despite the

intrinsic oxidation resistance provided by Mg, accumulation of water under prolonged industrial operation could still pose challenges to catalyst stability. Therefore, complementary reactor-level engineering strategies may be required for industrial applications. Specifically, strategies such as membrane-assisted in-situ water removal, downstream condensation coupled with recycle loops, and structured or hydrophobic reactor designs to reduce water retention have been successfully applied in catalytic processes. Integrating these reactor-level approaches with the intrinsic material-level oxidation resistance provided by Mg could yield synergistic improvements, significantly enhancing the practical viability of $Fe_7C_3$-based catalysts under industrially relevant conditions.

## Discussion

In summary, our study introduces a novel synthesis strategy for the $Fe_7C_3$ phase that leverages the unique roles of K and Mg promoters. By extracting $Fe_7C_3$ seeds from Prussian blue and employing K to enhance carbonization while using Mg to mitigate water-induced oxidation, we have successfully stabilized the $Fe_7C_3$ phase under demanding $CO_2$ hydrogenation conditions. Our $Fe_7C_3$-KMg catalyst exhibits catalytic performance that far exceeds that of conventional iron catalysts and opens up new avenues for the innovative design of $CO_2$ hydrogenation catalysts. Moreover, we have not only developed an efficient method to synthesize single-phase $Fe_7C_3$ but also elucidated the intrinsic link between the structure of $Fe_7C_3$ and its catalytic performance, thereby providing a solid foundation for further catalyst optimization. Future work will focus on optimizing this strategy further and exploring its applicability across a broader range of reaction conditions, paving the way for more sustainable and efficient $CO_2$ utilization technologies.

## Methods
### Catalyst preparation

All catalysts containing different promoters were synthesized by a coprecipitation method, using $Fe(NO_3)_3 \cdot 9H_2O$ as the iron source, and either $(NH_4)_4Fe(CN)_6$ or $K_4[Fe(CN)_6]$ as the precipitating agent. A suitable amount of polyvinylpyrrolidone K30 (average molecular weight ~40,000; TCI, Shanghai) was dissolved in 200 g of deionized water to ensure uniform dispersion and controlled nucleation of the PBA precursors. Subsequently, a solution of either $(NH_4)_4Fe(CN)_6$ (Honeywell Specialty Chemicals Seelze GmbH) or $K_4[Fe(CN)_6]$ (solution A) was prepared. Separately, $Fe(NO_3)_3 \cdot 9H_2O$ (AR, Sinopharm Chemical Reagent Co., Ltd.) was dissolved in 200 g of deionized water (solution B). Solution B was then slowly added into solution A under vigorous stirring to form the catalyst precursor. Finally, different amounts of $Mg(NO_3)_2 \cdot 4H_2O$ were introduced via incipient wetness impregnation. The final Fe/Mg ratio of each sample was determined by inductively coupled plasma-optical emission spectrometry (ICP-OES, Perkin Elmer). Catalyst samples are named based on the promoters employed: Fe refers to the iron catalyst without any promoter; FeMg refers to the catalyst modified with magnesium only; FeK denotes the catalyst promoted solely with potassium; and FeKMg indicates the catalyst co-promoted with both potassium and magnesium.

Specifically, the general designation "FeKMg" is used throughout the manuscript to describe the co-promoted catalyst. The notation "$Fe_7C_3$-KMg" is explicitly employed only when the presence of the $Fe_7C_3$ carbide phase has been confirmed through structural characterization.

### Catalyst characterizations

XRD patterns were acquired using a Bruker D8 powder diffractometer located in Karlsruhe, Germany, with Co Kα radiation (λ = 0.179 nm). The instrument operated at a voltage of 35 kV and a current of 40 mA. A continuous scan mode was employed with a step size of 0.04° and a dwell time of 0.4 seconds, covering a 2θ range from 40° to 75°.

TEM analysis was performed using an FEI Talos F200A electron microscope operating at 200 kV. The samples were sonicated in ethanol, then deposited onto a copper grid with a porous carbon film. Before testing, the samples were irradiated with an infrared lamp for 15 minutes to remove any residual solvents.

HAADF-STEM was used to capture STEM images (2048 × 2048 pixels) with a camera length of 260 mm and a spot diameter of 0.5 nm.

XPS spectra were recorded using a Thermo Scientific K-alpha system with Al Kα radiation (hν = 1486.6 eV) as the X-ray source. To prevent oxidation, the samples were prepared in a glove box. The C 1s peak (284.6 eV) was used as a reference for calibration.

PTH experiments for $C_3H_6$ were conducted using an AMI-300 apparatus equipped with a mass spectrometer. The catalysts were activated under ammonia gas, then switched to a 10% $H_2$/Ar flow (50 mL/min), with the temperature set to 340 °C. $C_3H_6$ was pulsed into the system to complete the PTH. The effluent was monitored for $C_3H_6$ (m/z = 42) and $C_3H_8$ (m/z = 44) using a PFEIFFER Omnistar mass spectrometer.

XAFS data were collected at the BL14W1 beamline of the Shanghai Synchrotron Radiation Facility (SSRF), China, operating at 3.5 GeV with a maximum current of 260 mA. Energy calibration was performed using the absorption edge of pure Fe foil, and XAFS data were acquired in fluorescence mode.

Mössbauer experiments were conducted using an MR-351 constant acceleration transmission spectrometer at 10 K with 25 mCi 57Co in a Rh matrix. Phase composition was identified based on isomer shift (IS), quadrupole splitting (QS), and magnetic hyperfine field ($H_{hf}$) parameters. The content of each phase was determined from the absorption peak areas, assuming the same recoil-free factor for all types of iron nuclei in the catalyst.

### Theoretical calculations

For calculation models, a periodical (1×1) of $h$-$Fe_7C_3$(211) slab was truncated from the optimized bulk phase, which contains 56 Fe and 23 C atoms. During the structural optimization, the bottom 27 Fe and 11 C atoms were fixed in the equilibrium positions as in the bulk phase while the others were allowed to relax. We selected the (211) facet for DFT modeling based on prior studies showing this to be a relatively low-index, catalytically active surface with moderate surface energy[32,33]. Additionally, the (211) plane exposes coordinatively unsaturated Fe atoms, suitable for $CO_2$/$H_2$ adsorption and promoter interaction modeling. A $MgCO_3$ cluster was loaded on the $h$-$Fe_7C_3$(211) facet to represent the Mg-promoted catalyst. All spin-polarized calculations were carried out with VASP code[34,35]. The frozen-core projector-augmented wave (PAW)[36] pseudo-potential with a cutoff energy of 450 eV was selected for the plane-wave expansion. The generalized gradient approximation in the Perdew-Burke-Ernzerhof (GGA-PBE)[37] with van deer Waals correction (D3)[38] was employed to describe the exchange-correlation energy. The convergence criteria for the force and electronic self-consistent iteration were set to 0.03 eV/Å and $10^{-5}$ eV, respectively. Gamma-centered (2×2×1) k-point was used for sampling of Brillouin zone. In all calculations, adsorption energies ($E_{ads}$) were calculated based on $E_{ads} = E_{x/slab} - [E_{slab} + E_x]$, where $E_{x/slab}$ is the total energy of the slab with adsorbents after full relaxation, $E_{slab}$ is the total energy of the bare slab, and $E_x$ is the total energy of the free adsorbents in the gas phase. Therefore, the more negative the $E_{ads}$, the stronger the adsorption. Reaction energies (ΔE) were defined as ΔE = $E_{final}$ - $E_{initial}$, where $E_{final}$ and $E_{initial}$ represent the final state energy and initial state energy, respectively. Therefore, a negative ΔE represents an exothermic process. All transition states were calculated using the climbing image nudged elastic band method (CI-NEB)[39], with the stretching frequencies analyzed in order to characterize whether a stationary point is a transition state with only one imaginary frequency.

## Catalyst testing

The catalytic efficiency of synthesized materials was systematically assessed using a quad-channel fixed-bed microreactor system. For each test run, 100 mg of catalyst was uniformly mixed with 200 mg quartz sand and packed into a quartz reaction chamber (10 mm ID) equipped with a temperature-monitoring stainless-steel sleeve. Standard evaluation parameters were maintained at 340 °C, 2.0 MPa, $H_2/CO_2 = 3$, and 6000 mL·$g_{cat}^{-1}$·$h^{-1}$ unless otherwise specified. Post-reaction products underwent phase separation through sequential hot (160 °C) and cold (0 °C) trapping systems, enabling collection of solid waxes, liquid hydrocarbons, and aqueous phases for offline analysis via HP-PONA 19091s-001 chromatography.

Continuous gas monitoring was achieved through Agilent 7890B GC with specialized detection modules: Gaspro-FID assemblies resolved C1-C4 hydrocarbons, while a PONA-FID system characterized C4-C7 compounds. Gas composition of $H_2/CO_2/CO/Ar$ was determined using coupled PLOT/Q, 5 Å molecular sieve, and Haysep Q columns interfaced with TCD detection. Carbon-based mass balance calculations confirmed experimental accuracy within ± 5% deviation through systematic comparison of inlet/outlet carbon fluxes.

The $CO_2$ conversion ($X_{CO2}$), product selectivity ($S_i$), reaction rate (R) and $C_{2+}$ olefin selectivity ($S_{C_{2+}^{=}}$) were calculated by the following equations:

$$X_{CO_2} = \frac{CO_{2in}/Ar_{in} - CO_{2out}/Ar_{out}}{CO_{2in}/Ar_{in}} \times 100\% \qquad (1)$$

$$S_i = \frac{N_i \times n_i}{\sum(N_i \times n_i)} \times 100\% \qquad (2)$$

$$R_{CO_2} = \frac{GHSV \times X_{CO_2} \times C_{CO_2}}{22400} \times 100\% \qquad (3)$$

$$S_{C_{2+}^{=}} = \frac{\sum S_i^{=}}{\sum(N_i \times n_i)} \times 100\% \qquad (4)$$

where $CO_{2\,in/out}$ denote molar flows of carbon dioxide at reactor feed and effluent streams respectively, $Ar_{in/out}$ refer to molar flows of argon at the reactor inlet and outlet. $S_i$ represents carbon-specific selectivity for product i, $N_i$ indicates molar fraction, $n_i$ corresponds to carbon count per molecule and $S_i^{=}$ is the specifically quantifies olefinic selectivity for i-carbon unsaturated hydrocarbons. GHSV corresponds to gas hourly space velocity. This analytical framework ensures comprehensive characterization of catalytic behavior while maintaining strict adherence to standardized evaluation protocols in heterogeneous catalysis research.

## Data availability

The data that support the findings of this study are available within the paper and its Supplementary Information, and all data are also available from the corresponding authors upon request. Source data are provided with this paper.

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

## Acknowledgements

The authors acknowledge financial support from the National Science Fund for Distinguished Young Scholars (22225206 to X.W. and 22025804 to Y.Y.), National Natural Science Foundation of China (22172183 to X.L., 22202224 to J.L. and 22178367 to G.L.), National Key R&D Program of China (2022YFA1604100 to X.W. and 2022YFB4101200 to Y.Y.), Beijing Natural Science Foundation (L255001 to X.L.), CAS Project for Young Scientists in Basic Research (YSBR-005 to X.W.), Key Research Program of Frontier Sciences CAS (ZDBS-LY-7007 to X.W.), National Natural Science Foundation Major Research Plan (92045303 to X.W.), CAS Informatization Plan (CAS-WX2021SF0110 to X.W.), Ordos Key R&D Program (YF20232316 to X.L. and YF20232317 to X.W.), and the funding support from Synfuels China, Co. Ltd.

## Author contributions

Y.Y., X.W. and X.L. designed the study. F.Q. performed most of the reactions and sample characterization. M.W. carried out the X-ray structure characterization and analysis. Y.C. performed the electron microscopy characterization. J.L. performed the DFT calculations. F.Q., X.L., J.L. Z.W., Z.S., R.L., M.Q., H.W., and G.L. wrote and revised the paper. All the authors discussed the results and commented on the manuscript.

## Competing interests

The authors declare no competing interests.
