## [Transparent Peer Review file · Nature Communications]

Stabilized Fe₇C₃ Catalyst with K–Mg Dual Promotion for Robust CO₂ Hydrogenation to High-Value Olefins

Corresponding Author: Professor Xiao-Dong Wen

Version 0:

Reviewer comments:

Reviewer #1

(Remarks to the Author)

CO₂ hydrogenation to high value olefins, as one of important routes for CO₂ utilization and conversion, is attracting much more attention. The studies on the design and preparation of its efficient catalysts are still its hot-spot researches. In this manuscript, Fei Qian et al. designed a good catalyst of Fe₇C₃-KMg, which exhibits good catalytic performances for CO₂ hydrogenation to olefins: 41.5% of CO₂ conversion and 67.1% of olefins selectivity could be kept over 400h at 340°C, 2MPa, and H₂/CO₂=3. Simultaneously, the roles of each active components and contributions to the reactions are elucidated. The conclusions are helpful to the catalyst design for CO₂ hydrogenation. However, due to the following comments, it is thought that this manuscript is not suitable for publishing at current stage.

The strategic preparation of iron-based catalysts with co-promoters of alkali-metal and alkali-earth metal have ever been reported in literatures, like NaSrFe, NaBaFe KBaFe catalysts, etc., which also exhibits the good catalytic performances for CO₂ hydrogenation to olefins. E.g. NaSrFe catalysts exhibit good catalytic performances for CO₂ to olefins: 40.5% of CO₂ conversion and 77.5% of C₂₊ olefins selectivity could be obtained at 320°C, 3.0MPa, and H₂/CO₂=3 and also exhibit excellent catalytic stability during 500h stability experiments. (Applied Catalysis B: Environmental, 316(2022)121640.) Such results should be added to the compared results of literature results and have a discussion about the differences among them to this paper, and further strengthen the novelty of this manuscript.

Reviewer #2

(Remarks to the Author)

The manuscript presents a detailed study on the development of a nearly phase-pure Fe₇C₃ catalyst promoted by potassium (K) and magnesium (Mg) for CO₂ hydrogenation to high-value olefins. The authors demonstrate that the Fe₇C₃ phase, derived from Prussian blue analogues (PBAs) and stabilized through strategic K and Mg promotion, achieves high CO₂ conversion (41.5%) and olefin selectivity (67.1%) with remarkable stability over 400 hours. The work is well-supported by comprehensive characterization (XRD, Mössbauer, XAS, XPS, TEM, etc.) and theoretical calculations (DFT), providing insights into the roles of K and Mg in enhancing carbonization and suppressing water-induced oxidation, respectively.

Below, I outline specific comments and questions, incorporating additional points for improvement.

1. The manuscript identifies the stabilization of Fe₇C₃ as a major challenge, citing thermodynamic and kinetic constraints, complex formation pathways, and susceptibility to oxidation under high water partial pressures (Page 4, Lines 71-79). These points are well-articulated, but the discussion could be expanded to address practical implications for industrial applications. For instance:

o Thermodynamic Stability: The authors note that Fe₇C₃ formation is less favored compared to Fe₅C₂ or metallic Fe under typical CO/H₂ ratios. Could the authors quantify the thermodynamic window (e.g., temperature, pressure, gas composition) where Fe₇C₃ is stable? A phase diagram or reference to computational studies would enhance this discussion.

o Pretreatment Conditions: The use of NH₃ activation to form Fe₂N as a precursor is innovative, but the manuscript lacks details on the scalability of this process. Are there energy or safety concerns associated with NH₃ pretreatment at 550°C (Page 3, Supplementary Fig. 2)? How does this compare to conventional CO/H₂ carburization methods?

o Oxidation Resistance: The role of Mg in suppressing water-induced oxidation is compelling, but the manuscript does not discuss how this might translate to real-world conditions with fluctuating CO₂/H₂ ratios or impurities (e.g., O₂, H₂S). Could the

authors comment on the robustness of Fe₇C₃-KMg under non-ideal feed conditions?

Expand the discussion in the Introduction or Discussion section to include these practical considerations, potentially referencing industrial FTS processes or citing relevant literature.

2. I want to raise a critical point about the redox nature of CO₂ hydrogenation, where reduction (CO₂ to hydrocarbons) is coupled with oxidation reactions, potentially re-oxidizing Fe to Fe_xO_y (e.g., Fe₃O₄). The authors demonstrate that Mg suppresses water dissociation, reducing oxidation of Fe₇C₃ (Pages 10-12, Fig. 4, Fig. 5). However, the manuscript does not fully address system-level strategies to mitigate oxidation beyond Mg promotion. Specific concerns include:

o Water Management: In FTS, water is a major byproduct that can oxidize iron carbides. The authors show that Mg reduces water dissociation (Fig. 5a), but could system design strategies—such as water removal via membranes, condensers, or recycle loops—complement Mg's role? A brief discussion of reactor engineering solutions would strengthen the manuscript's practical relevance.

o Fe_xO_y/Fe₇C₃ Synergy: I am curious whether a mixed Fe_xO_y/Fe₇C₃ system might be preferable. Table 1 shows that FeK (46% Fe₃O₄, 53% Fe₇C₃) achieves reasonable CO₂ conversion (35.9%) and olefin selectivity (49.2%), suggesting that Fe₃O₄ may contribute to reverse water-gas shift (RWGS) activity, as noted in the literature (Page 3, Lines 46-51). However, FeKMg (98% Fe₇C₃) outperforms FeK, implying that Fe₇C₃ is the primary active phase for olefin production. Could the authors clarify whether trace Fe₃O₄ in FeKMg (not detected in XRD but possibly present below detection limits) plays a synergistic role? Alternatively, is a phase-pure Fe₇C₃ system optimal, or could controlled Fe₃O₄ incorporation enhance RWGS while maintaining Fe₇C₃ stability?

o Promoter Interactions: The manuscript attributes K to carbonization and Mg to oxidation resistance, but how do these promoters interact with the reaction environment (e.g., H₂O, CO, H₂) to prevent oxidation? For instance, does K influence water adsorption, or is its role limited to electronic effects on Fe₇C₃?

Add a paragraph in the Discussion section addressing system-level strategies (e.g., water management) and the potential role of Fe₃O₄/Fe₇C₃ synergy. Conduct additional experiments or cite literature to clarify whether trace Fe₃O₄ is beneficial or detrimental. If Fe₃O₄ is absent in FeKMg, explicitly state this based on characterization limits (e.g., Mössbauer sensitivity).

3. Figure 1i (line scan) shows uniform distribution of K and Mg in FeKMg particles, but the manuscript does not clarify their structural integration. Supplementary Fig. 14 indicates that Mg is enriched on the surface as MgCO₃, while K's form is less clear. Specific questions include:

o K and Mg in Fe₇C₃ Lattice: Are K and Mg incorporated into the Fe₇C₃ lattice (e.g., as dopants or interstitials), or do they exist as separate phases (e.g., K₂O, MgCO₃)? XPS suggests MgCO₃ (Supplementary Fig. 15), but K's chemical state is not discussed. Could the authors provide XPS or XANES data for K to confirm its form (e.g., K₂O, K₂CO₃, or metallic K)?

o Alloying vs. Phase Separation: Whether K and Mg form alloys or separate phases. STEM-EDS (Fig. 1h-i) suggests homogeneous distribution, but MgCO₃ detection implies phase separation for Mg. DFT calculations model Mg as a MgCO₃ cluster on Fe₇C₃ (Page 16, Lines 369-373), supporting a surface phase. For K, the manuscript suggests electron donation to Fe₇C₃ (Supplementary Fig. 22), but its structural role (e.g., surface oxide vs. lattice incorporation) remains ambiguous. Clarifying this would strengthen the mechanistic claims.

o Interaction Between K and Mg: It is noted that K and Mg's effects (carbonization and stabilization, respectively) are achieved together (Fig. 2). Is there evidence of K-Mg interaction (e.g., K-modifying MgCO₃ stability or Mg influencing K's electronic effects)? The DFT model (Fig. 5a) treats Fe₇C₃-K and Fe₇C₃-KMg separately, but a combined K-Mg model could reveal synergistic effects.

Include XPS or XANES data for K to clarify its chemical state. Discuss whether K and Mg are lattice-incorporated or phase-separated, referencing STEM-EDS and XPS results. If feasible, extend DFT calculations to model K-Mg interactions on Fe₇C₃ to confirm synergy or independence. Revise Fig. 2f to reflect these structural details.

4. Table 1 shows that FeKMg (41.5% CO₂ conversion, 98% Fe₇C₃) outperforms Fe (33.7%, 19% Fe₇C₃), but the improvement is only ~1.23-fold, which I find modest. Additionally, C₁-CO selectivity is comparable (9.1% for Fe vs. 12.4% for FeKMg). These points raise questions about FeKMg's superiority:

o Modest Conversion Increase: The ~1.23-fold increase in CO₂ conversion is notable but not dramatic, possibly due to FeKMg's high Fe₇C₃ purity optimizing FTS over RWGS, while Fe's Fe₃O₄ dominance favors RWGS (higher CO selectivity). However, FeKMg's olefin selectivity (67.1%) is ~2.8 times higher than Fe's (23.9%), which is a significant improvement. The manuscript should emphasize this selectivity advantage over conversion to highlight Fe₇C₃'s value. Could the authors discuss whether conversion is limited by thermodynamics, kinetics, or mass transfer (e.g., pore blocking by MgCO₃)?

o C₁-CO Selectivity: The similar CO selectivity (9.1% vs. 12.4%) suggests that FeKMg retains some RWGS activity, possibly from minor Fe₃O₄ or K-promoted sites. Supplementary Fig. 10 shows FeKMg's CO selectivity stabilizes at ~12%, while Fe's increases over time, indicating better FTS dominance in FeKMg. Could the authors quantify the RWGS contribution in FeKMg (e.g., via CO yield or isotopic studies) to explain this?

o Comparison to Literature: Supplementary Table 8 compares Fe₇C₃-K-0.04Mg (67.1% olefin selectivity at 6000 mL·g⁻¹·h⁻¹) to other catalysts, showing superior performance. However, the manuscript does not discuss why FeKMg outperforms alternatives (e.g., NaFe, ZnFeAlO₄). Is it solely due to Fe₇C₃ purity, or do K-Mg promoters provide unique electronic effects? In the Discussion, emphasize FeKMg's olefin selectivity advantage over conversion to justify its superiority. Analyze potential limitations to conversion (e.g., thermodynamics, diffusion). Clarify RWGS contributions in FeKMg using existing data or new experiments (e.g., CO yield vs. time). Expand the comparison in Fig. 3d to highlight why FeKMg excels (e.g., phase purity, promoter effects).

5. The manuscript states that Mg is present as MgCO₃ on the catalyst surface (Supplementary Fig. 14-15) and suppresses water-induced oxidation (Fig. 4, Fig. 5). It is important to know the cause of MgCO₃ formation and its mechanism for oxidation resistance:

o MgCO₃ Formation: MgCO₃ likely forms via reaction of Mg²⁺ with CO₂ or CO under reaction conditions (340°C, 2 MPa,

H₂/CO₂=3), as CO₂ is abundant and MgO is prone to carbonation. The manuscript confirms this via XPS (Supplementary Fig. 15), but does not discuss why Mg remains as MgCO₃ rather than MgO or metallic Mg. Is MgCO₃ thermodynamically favored under these conditions? A brief thermodynamic analysis or reference to Mg-CO₂ interactions would clarify this.

- o Oxidation Suppression Mechanism: Fig. 5a shows that Mg increases the energy barrier for the second O-H dissociation of H₂O (0.84 eV vs. 0.66 eV for Fe₇C₃-K), reducing oxygen adsorption. COHP analysis (Fig. 5b) indicates weaker Fe-O bonds with Mg, and Supplementary Fig. 22 suggests MgCO₃ accepts electrons, creating a charge-deficient Fe₇C₃ surface less prone to oxidation. However, it's unclear how MgCO₃—a surface species—physically or chemically shields Fe₇C₃. Does MgCO₃ act as a hydrophobic barrier, an electron sink, or a site-blocking layer? The D₂O-TPD data (Supplementary Fig. 17) show lower D₂ signals for FeKMg, supporting reduced dissociation, but could MgCO₃ also adsorb water, preventing it from reaching Fe₇C₃?
- o Stability of MgCO₃: Under high water partial pressures, MgCO₃ could hydrolyze to Mg(OH)₂, yet FeKMg remains stable (Fig. 4a). How does MgCO₃ resist transformation, and does it regenerate during reaction (e.g., via CO₂)? Clarify MgCO₃ formation by discussing its thermodynamics or citing relevant studies (e.g., MgO carbonation in CO₂ capture). Elaborate on MgCO₃'s mechanism in the Discussion, specifying whether it blocks water access, weakens Fe-O bonds, or both. If possible, add experiments (e.g., Mg 1s XPS after water exposure) to confirm MgCO₃'s stability. Revise Fig. 5c to illustrate MgCO₃'s role more explicitly.

Some Minor Comments:

- o Fig. 2e quantifies phase evolution, but the transition from Fe₂N to Fe₇C₃ in FeKMg is rapid (within 8 h). Could the authors comment on the kinetics of this transformation? Is Fe₂N a critical intermediate, or could other precursors achieve similar results?
- o The schematic in Fig. 2f is helpful but oversimplifies K and Mg's roles. Consider adding MgCO₃ and K's likely form (e.g., K₂O) to reflect characterization data.
- o Page 15 describes coprecipitation with Fe(NO₃)₃ and Mg(NO₃)₂, but the role of polyvinylpyrrolidone (PVP) is unclear. Is PVP a stabilizer, and does it affect Fe₇C₃ formation? Clarify its necessity or cite relevant PBA synthesis protocols.
- o The impregnation of Mg is mentioned briefly (Line 332). Specify the Mg loading method (e.g., wet impregnation, incipient wetness) and its impact on MgCO₃ distribution.
- o The DFT model uses a MgCO₃ cluster on Fe₇C₃ (211) (Page 16). Why was this facet chosen, and how sensitive are the results to surface orientation? A brief justification or reference to Fe₇C₃'s active facets would help.
- o The absence of K-Mg interaction in DFT is a limitation. If computational constraints prevent this, acknowledge it in the Methods or Discussion.
- o Supplementary Table 8 lists 24 catalysts, but the discussion in Fig. 3d is brief. Highlight key differences (e.g., Fe₇C₃ vs. Fe₅C₂, promoter effects) to contextualize FeKMg's performance.
- o Reference 19 (Pasupulety et al.) reports Fe₇C₃ in FeZnK/ZrO₂ with lower performance. Explicitly compare this to FeKMg to underscore the advantage of PBA-derived Fe₇C₃.
- o Supplementary Fig. 9 and Fig. 3a-b use pie charts for olefin fractions, which are less precise than bar charts. Consider replacing pie charts with quantitative breakdowns (e.g., C₂⁼, C₃⁼, C₄⁼ selectivities).
- o Supplementary Table 1 and Table 5 have overlapping Mössbauer data for FeKMg. Consolidate or clarify why both are needed.
- o "High-value olefins" is used throughout but not defined. Specify which olefins (e.g., C₂⁼-C₄⁼) are targeted, as C₅⁺ olefins may also be valuable in FTS.
- o "Phase-pure" Fe₇C₃ is claimed (e.g., Page 9, Line 204), but Table 1 lists 98% Fe₇C₃ for FeKMg. Use "nearly phase-pure" consistently to avoid overstatement.

The manuscript presents a significant advance in CO₂ hydrogenation catalysis by demonstrating a stable, selective Fe₇C₃-based catalyst with K-Mg promotion. The combination of experimental characterization, catalytic testing, and DFT calculations provides robust evidence for Fe₇C₃'s potential and the roles of K and Mg. The work is novel, addressing a gap in Fe₇C₃-focused studies, and the results are compelling, particularly the high olefin selectivity and 400-hour stability. However, the manuscript would benefit from addressing the reviewer's questions and my additional comments to enhance clarity, mechanistic insight, and practical relevance. Specifically, clarifying K and Mg's structural roles, discussing system-level oxidation prevention, and justifying FeKMg's performance advantages are critical. With these revisions, the manuscript will be suitable for publication in Nature Communications.

Reviewer #3

(Remarks to the Author)

General comments: This article presents a Fe₇C₃-based catalytic system for CO₂ hydrogenation to olefins. The authors successfully synthesized Fe₇C₃ phase using a Prussian blue molecular precursor. The olefin yield was increased by using two promoters. Potassium (K) inhibits the secondary hydrogenation, providing high olefin selectivity, and magnesium (Mg) improves the water deactivation resistance of the catalyst. Combining catalytic performance evaluation, in situ/ex situ characterization, and DFT calculations, the authors provided mechanistic insights into the role of each promoter and demonstrated the long-term stability and excellent selectivity of the Fe₇C₃-K-Mg catalyst. As mentioned by the authors of this paper, Fe-carbide phases were prepared and characterized using various approaches as active species for CO₂ hydrogenation. Fe₇C₃ and Fe₅C₂ were identified in the CO₂ hydrogenation reaction, where the initial iron oxide undergoes a phase transition in the reducing environment in the presence of water. Since RWGS converts CO₂ to CO, the original iron oxide or metallic Fe species still exists along with the resulting Fe-carbide phase. Many previous literatures have investigated the relative amounts of Fe-carbide and Fe-oxide depending on the reaction conditions and the types of alkaline

promoters (e.g., Na, K, Mg) and transition metal promoters (e.g., Co, Mn, Zn, Cu). In most cases, mixed carbide phases inevitably appear due to various phase transitions in the presence of CO, H₂, and water at high temperature (~320 °C) and high pressure. Due to these limited environments and the presence of various trace oxidizing and reducing agents, the exact phase transitions due to the promoter effect have rarely been elucidated. Pure Fe-carbide phases derived from the initial Fe-oxide phase have rarely been reported. Even using synchrotron-based in situ characterization, it was impossible to identify pure carbide species reflecting high pressure conditions. Even using XRD, the relative iron oxide or metallic peaks were so dominant that it was difficult to clearly identify the carbide peaks. Various combinations of alkaline or transition metal promoters and Fe₂O₃ or Fe₃O₄ resulted in changes in CO₂ conversion and selectivity for methane, light olefins, aromatics and C₅+ hydrocarbons over a reasonable range of ASF distributions. There is no other paper beyond this type of study for CO₂ hydrogenation using RWGS and FTS routes. The authors provide the first clear evidence that Fe₇C₃ can act as a durable and highly active phase in CO₂ hydrogenation, but they followed the previous studies without any new characterization techniques or new discoveries of new promoters or new species. This suggests that the formation of Fe₂N on fresh Prussian Blue-based catalysts can promote the formation of pure Fe₇C₃ phase, which is a unique feature of this study. However, there is no proper evidence, mechanism or correlation between Fe₂N and Fe₇C₃ during the phase transformation process. Why does Fe₂N transform into iron oxide and iron carbide phases? Why do the active species or product yields change so dramatically in the presence of K, Mg, and K-Mg on Fe-carbide catalyst? Can the authors be sure that there are no Fe-oxide or Fe-carbide species other than Fe₇C₃ in the spent catalysts? Why is that?

Most of the results of this study are based on existing knowledge in the literature. The authors provided a couple of evidences about pure Fe₇C₃ in this work. However, many areas remain unclear. Some promoters change different structures and catalytic reactivities, but the reasons are not clear. This paper does not meet the criteria for publication in a prestigious journal such as Nat. Comm., which requires new analysis, interpretation, or the discovery of new species. The reviewer considers this article interesting but not sufficient for publication in Nat Comm. Additional questions are given below.

1. While the Fe K-edge XANES results clearly demonstrate that Mg incorporation suppresses Fe oxidation under reaction conditions, the proposed electron-withdrawing effect of MgCO₃—suggested by DFT calculations—has not been directly validated experimentally. In particular, the manuscript lacks spectroscopic evidence that would confirm electron transfer from Fe₇C₃ to Mg, such as Mg K-edge XANES. Including such analyses would help distinguish whether the observed stabilization of Fe₇C₃ arises from electronic modulation or simply from a physical barrier effect due to surface MgCO₃.
2. For the XAFS analysis, especially EXAFS, reference data for iron carbides (Fe₃C₂ or Fe₇C₃) are necessary. Since EXAFS analysis involves comparing the types and coordination numbers of atoms surrounding Fe, reference data related to Fe–C bond should be provided to confirm the formation of iron carbides in the spent iron catalysts.
3. ICP results should be provided to confirm the K and Mg contents of iron catalysts. Especially, in the catalytic characterization and activity results shown in Figs. 1–4 and Table 2, Mg content for each catalyst (FeMg, FeKMg) should be provided.

Minor comments

1. The reaction time for CO₂ hydrogenation over the four spent catalysts (Fe, FeMg, FeK, and FeKMg), as indicated in Fig. 1 for the characterization of spent iron catalysts, should be explicitly described in the main text.
2. Consistent sample naming is required (e.g., FeKMg vs. Fe₇C₃-KMg, except for DFT result). In addition, a clear explanation of the sample nomenclature should be provided in the main text or “Catalyst preparation” section.
3. In the figure legend of Fig. 3(c), there is a typo “CO conversion”, and it should be corrected to “CO₂ conversion”.

Version 1:

Reviewer comments:

Reviewer #1

(Remarks to the Author)

The authors carefully replied to the reviewers' comments and improved the quality of the manuscript, I think it meet the standards of being published and recommend it to be accepted for publication.

Reviewer #3

(Remarks to the Author)

This manuscript has been substantially upgraded after including sufficient response to reviewer comments. This manuscript is considered suitable for publication in the journal.

Response to Reviewers:

We sincerely thank the reviewers for their insightful and constructive comments, which have significantly enhanced the quality of our manuscript. We have carefully considered and addressed each comment with a detailed, point-to-point response. The manuscript has been thoroughly revised according to the reviewers' suggestions. In this response letter, the reviewers' original comments are presented in black italics, our point-to-point responses are provided in blue, and all modifications made in the revised manuscript and supporting information are highlighted in red.

REVIEWER COMMENTS

Reviewer #1 (Remarks to the Author):

CO₂ hydrogenation to high value olefins, as one of important routes for CO₂ utilization and conversion, is attracting much more attention. The studies on the design and preparation of its efficient catalysts are still its hot-spot researches. In this manuscript, Fei Qian et al. designed a good catalyst of Fe₇C₃-KMg, which exhibits good catalytic performances for CO₂ hydrogenation to olefins: 41.5% of CO₂ conversion and 67.1% of olefins selectivity could be kept over 400 h at 340°C, 2 MPa, and H₂/CO₂=3. Simultaneously, the roles of each active components and contributions to the reactions are elucidated. The conclusions are helpful to the catalyst design for CO₂ hydrogenation. However, due to the following comments, it is thought that this manuscript is not suitable for publishing at current stage.

The strategic preparation of iron-based catalysts with co-promoters of alkali-metal and alkali-earth metal have ever been reported in literatures, like NaSrFe, NaBaFe KBaFe catalysts, etc., which also exhibits the good catalytic performances for CO₂ hydrogenation to olefins. E.g. NaSrFe catalysts exhibit good catalytic performances for CO₂ to olefins: 40.5% of CO₂ conversion and 77.5% of C₂+ olefins selectivity could be obtained at 320°C, 3.0 MPa, and H₂/CO₂=3 and also exhibit excellent catalytic stability during 500 h stability experiments. (Applied Catalysis B: Environmental,

316(2022)121640.) Such results should be added to the compared results of literature results and have a discussion about the differences among them to this paper, and further strengthen the novelty of this manuscript.

Author reply: We sincerely appreciate the reviewer's insightful and constructive comments, which have greatly improved our manuscript. As the reviewer accurately pointed out, previously reported Fe-based catalysts promoted by alkali and alkaline-earth metals, such as NaSrFe, NaBaFe, and KBaFe (especially the impressive work published in *Appl Catal. B: Environ.* **316**, 121640 (2022)), have indeed shown excellent catalytic performance. We have carefully cited this key reference, searched and added a detailed comparative discussion in the revised manuscript, clearly highlighting the differences and unique aspects of our study. We respectfully emphasize the distinctive contributions of our work compared to previous studies:

(1) First demonstration of precise synthesis and robust stabilization of nearly phase-pure Fe₇C₃: Previous studies predominantly rely on dynamic equilibria between χ -Fe₅C₂ and Fe₃O₄, with χ -Fe₅C₂ identified as the dominant active phase (*Science* **387**, eadn9388 (2025); *Nat. Synth.* **4**, 288-302 (2025)). In contrast, our work presents, for the first time, a synthetic strategy enabling reliable preparation and long-term stabilization (>1000 h see **Fig. R1**) of nearly phase-pure Fe₇C₃ under realistic CO₂ hydrogenation conditions. This achievement addresses a long-standing challenge, establishing a fundamentally new paradigm for catalyst design.

(2) Exceptional structural stability and catalytic performance under milder reaction conditions: Unlike previously reported systems, which often undergo gradual oxidation and formation of inactive Fe₃O₄ phases, our Fe₇C₃-KMg catalyst demonstrates outstanding structural integrity and catalytic stability (>1000 h) without detectable oxidation. Furthermore, it achieves comparable catalytic performance (CO₂ conversion of 41.5% and olefin selectivity of 67.1%) at a lower and more economically favorable pressure (2.0 MPa) compared to previous catalysts requiring higher pressures (e.g., 3.0 MPa).

(3) Clear mechanistic elucidation of the dual K–Mg promotion: Comprehensive

experimental characterization combined with detailed DFT calculations explicitly identifies the distinct but complementary roles of K and Mg promoters: K accelerates catalyst carburization and enhances olefin selectivity, while Mg provides robust oxidation resistance by modulating the electronic environment of Fe_7C_3 . This clear mechanistic insight significantly advances the fundamental understanding of Fe-based catalyst systems.

Fig. R1 Stability test of the Fe_7C_3 -KMg catalyst. (Reaction conditions: 0.10 g of catalyst, 613 K, 2.0 MPa, $\text{H}_2/\text{CO}_2 = 3$, GHSV = $6 \text{ L} \cdot \text{g}_{\text{cat}}^{-1} \cdot \text{h}^{-1}$).

Following the reviewer's suggestion, we have explicitly summarized these comparative advantages in the revised manuscript (*see Page 8 Line 171 – 180, Page 10-11 Line 229 – 236*), clearly highlighting the novelty and practical significance of our catalyst system relative to existing literature.

Additionally, to thoroughly address the reviewer's concerns regarding the manuscript's initial suitability for publication, we have extensively revised our manuscript to enhance its rigor, clarity, and depth (please see *Page 7–8, Lines 159–180; Page 10–11, Lines 222–237; Page 11–12, Lines 249–259; Page 15, Lines 339–356*). We have also updated *Figures 1c, 1d, 3a, 3b, and 3e*, revised *Supplementary Figure 10*, and added *Supplementary Figures 9, 15, 17, 18, and 19*, as well as *Supplementary Tables 2 and 3*, further demonstrating our earnest efforts to improve manuscript quality comprehensively.

Lastly, although not explicitly raised by Reviewer #1, we respectfully emphasize that our identification of a single-phase Fe_7C_3 catalytic model—breaking the traditional χ - $\text{Fe}_5\text{C}_2/\text{Fe}_3\text{O}_4$ dual-phase paradigm—represents a scientifically significant advancement. This finding fundamentally challenges existing theories that regard Fe_3O_4 as

indispensable, introducing an entirely new catalytic concept in the field of iron-based CO₂ hydrogenation.

We sincerely thank the reviewer again for the valuable comments, which have significantly strengthened the manuscript and clarified its novelty and significance for publication in *Nature Communications*.

Reviewer #2 (Remarks to the Author):

The manuscript presents a detailed study on the development of a nearly phase-pure Fe₇C₃ catalyst promoted by potassium (K) and magnesium (Mg) for CO₂ hydrogenation to high-value olefins. The authors demonstrate that the Fe₇C₃ phase, derived from Prussian blue analogues (PBAs) and stabilized through strategic K and Mg promotion, achieves high CO₂ conversion (41.5%) and olefin selectivity (67.1%) with remarkable stability over 400 hours. The work is well-supported by comprehensive characterization (XRD, Mössbauer, XAS, XPS, TEM, etc.) and theoretical calculations (DFT), providing insights into the roles of K and Mg in enhancing carbonization and suppressing water-induced oxidation, respectively.

Author reply: We sincerely thank the reviewer for the thorough and insightful evaluation of our manuscript. We are especially grateful for recognizing our work's novelty, the robust integration of experimental characterization and DFT calculations, and the compelling catalytic performance (high selectivity and stability) achieved by our K-Mg promoted Fe₇C₃ catalyst. In response to the reviewer's valuable suggestions to further enhance clarity, mechanistic understanding, and practical relevance, we have explicitly expanded discussions on promoter roles, clarified the mechanisms underlying catalyst stability and oxidation resistance, and incorporated a detailed comparison with relevant literature and industrial scenarios. These specific improvements are now clearly reflected in the revised manuscript.

1. The manuscript identifies the stabilization of Fe₇C₃ as a major challenge, citing thermodynamic and kinetic constraints, complex formation pathways, and susceptibility to oxidation under high water partial pressures (Page 4, Lines 71-79). These points are well-articulated, but the discussion could be expanded to address practical implications for industrial applications. For instance:

o Thermodynamic Stability: The authors note that Fe₇C₃ formation is less favored compared to Fe₅C₂ or metallic Fe under typical CO/H₂ ratios. Could the authors

quantify the thermodynamic window (e.g., temperature, pressure, gas composition) where Fe₇C₃ is stable? A phase diagram or reference to computational studies would enhance this discussion.

Author Reply: We sincerely thank the reviewer for this valuable suggestion regarding the thermodynamic stability of Fe₇C₃ under typical FTS or CO₂ hydrogenation conditions. Indeed, understanding the stability windows of iron carbide phases has been a key aspect in the development and optimization of iron-based CO₂/CO hydrogenation catalysts.

However, due to the inherent complexity of the Fe–C phase system under realistic reaction conditions—such as dynamic shifts in carbon chemical potential, variable H₂/CO (or CO₂) ratios, and oxidation potential induced by water—the precise thermodynamic boundaries for Fe₇C₃ stability remain challenging to quantify with absolute accuracy. Although extensive theoretical and computational studies have been carried out (including works from our own group as well as other leading theoretical groups), strictly quantitative phase diagrams that pinpoint exact temperature, pressure, and gas-composition windows for Fe₇C₃ are still limited. Rather, existing computational results predominantly offer qualitative or semi-quantitative trends, which clearly indicate Fe₇C₃ formation is thermodynamically less favorable than other common iron carbides such as Fe₅C₂ or Fe₃C under typical reaction conditions.

From the experimental perspective, the observation of pure or near-pure Fe₇C₃ has been rarely reported, typically only under specific circumstances such as the presence of strong promoters or higher pressures and carbon potentials. Our group's prior work (*ACS Catal.* **8**, 3304-3316 (2018)) and Bert Weckhuysen's group (*J. Am. Chem. Soc.* **132**, 14928-14941 (2010)) have explicitly demonstrated through both experimental observations and theoretical analyses that Fe₇C₃ is a metastable phase, achievable only under conditions of high carbon chemical potential and controlled reaction environments. These studies collectively highlight that Fe₇C₃ generally exists within a narrow and somewhat extreme thermodynamic window (high carbon potential, moderate temperature, and pressure conditions).

To clarify this point, we have explicitly summarized the current state-of-the-art understanding of Fe₇C₃ stability in our revised manuscript, citing these crucial experimental and theoretical studies. While we acknowledge the challenges in constructing precise quantitative phase diagrams, we have made a concerted effort to explore this aspect in our study. In parallel, we emphasize the novelty of our catalyst design strategy, which utilizes a Prussian blue-derived Fe₂N intermediate in combination with K-Mg promoters to successfully stabilize Fe₇C₃ under industrially relevant reaction conditions.

As recommended, we have conducted theoretical calculations on the thermodynamic formation energies of various iron carbides as a function of carbon chemical potential (μ_C), with results presented in Figure R2a. The calculations reveal that two Fe₂C phases exhibit the lowest formation energy, followed by Fe₇C₃, Fe₅C₂, and Fe₃C. Specifically, when $\mu_C < -8.0$ eV, Fe₇C₃ and Fe₅C₂ show comparable thermodynamic stability, whereas for $\mu_C > -8.0$ eV, Fe₇C₃ demonstrates a lower formation energy than Fe₅C₂, indicating its superior thermodynamic stability. Notably, across the entire range of μ_C considered, the maximum formation energy difference between phases is within 0.1 eV/atom, suggesting a flat potential energy surface for iron carbides and thus facilitating facile phase transformations among them. In Figure R2b, we plot the contour of carbon chemical potential as a function of temperature and pressure. Here, μ_C is defined based on the disproportionation reaction ($2\text{CO} = \text{CO}_2 + \text{C}$), such that $\mu_C = 2\mu_{\text{CO}} - \mu_{\text{CO}_2}$. Higher temperatures and lower CO partial pressures result in lower carbon chemical potentials. It provides a direction for experiment to control synthesis desirable phases by modulating temperature and pressure. It is important to emphasize that while kinetic factors play a critical role in iron carbide phase transformations, they remain challenging to address in purely theoretical studies, representing a direction for future experimental-computational integration.

We greatly appreciate the reviewer's insightful comments, as this discussion significantly improves the clarity and practical relevance of our manuscript (*see Page 7-8 line159 – Page 8 170, Supplementary Fig. 9*).

Figure R2. (a) The calculated formation energy (eV/atom) of iron carbides with carbon chemical potential (μ_C). (b) The carbon chemical potential (μ_C) with temperature and partial pressure.

o Pretreatment Conditions: The use of NH₃ activation to form Fe₂N as a precursor is innovative, but the manuscript lacks details on the scalability of this process. Are there energy or safety concerns associated with NH₃ pretreatment at 550°C (Page 3, Supplementary Fig. 2)? How does this compare to conventional CO/H₂ carburization methods?

Author reply: We sincerely thank the reviewer for this valuable comment. We fully acknowledge that considerations regarding scalability, energy efficiency, and safety for the NH₃ pretreatment are crucial for potential industrial applications. The process was conducted under well-controlled conditions, with no signs of abnormal pressure buildup or other safety-related issues observed throughout the study. In addition, tail gases were properly treated via absorption, ensuring safe operation without environmental or safety concerns. With appropriate gas handling protocols, the NH₃ pretreatment is considered both safe and reliable. However, the primary focus of this manuscript is to explore and validate the scientific innovation and effectiveness of nitridation pretreatment as a precise strategy for controlling catalyst phase structures. In fact, the use of NH₃ nitridation pretreatment to prepare specific nitride precursors, thereby guiding the formation of target phases with high precision that direct

carburization process can't do, has multiple successful scientific precedents. For example, in our previous work, we successfully utilized NH_3 nitridation to generate a highly stable Fe_2N precursor, which subsequently facilitated the formation of Fe_2C with superior FTS catalytic performance (*Nat Commun*, **15**, 5128 (2024)). Similarly, Prof. Ding Ma's group employed an analogous NH_3 nitridation strategy to convert MoO_2 into a Mo_2N precursor at 700 °C, which was then precisely carburized using methane at 590 °C to preferentially form the highly active α -MoC phase, effectively avoiding the formation of the less active β -MoC phase (*Nature* **638**, 690–696 (2025)). This strategy has been well-established and widely recognized in the scientific community.

Therefore, the NH_3 pretreatment method employed in our study is supported by robust scientific rationale and successful prior demonstrations, underscoring its effectiveness in precisely controlling catalyst phase structures. Specific details and optimization for industrial-scale implementation, although important, extend beyond the scope of the current fundamental investigation and represent considerations for future development. Following the reviewer's valuable suggestion, we have explicitly added a brief discussion of these scalability, energy, and safety considerations into the Note of Supplementary Fig. 2 (*Page 3 in SI*), clearly acknowledging the importance of further optimizing these factors for future industrial-scale implementation.

We sincerely thank the reviewer again for this insightful feedback, which significantly improved our manuscript.

o Oxidation Resistance: The role of Mg in suppressing water-induced oxidation is compelling, but the manuscript does not discuss how this might translate to real-world conditions with fluctuating CO_2/H_2 ratios or impurities (e.g., O_2 , H_2S). Could the authors comment on the robustness of $\text{Fe}_7\text{C}_3\text{-KMg}$ under non-ideal feed conditions?

Expand the discussion in the Introduction or Discussion section to include these practical considerations, potentially referencing industrial FTS processes or citing relevant literature.

Author reply: We thank the reviewer for raising this important point regarding to the

robustness of Fe₇C₃-KMg under realistic, non-ideal feed conditions. To directly address this concern, we have conducted additional long-term stability tests under fluctuating CO₂/H₂ ratios (ranging from 1:1 to 4:1), closely simulating industrial feed variability.

Fig. R3 (a) Conversion of the Fe₇C₃-KMg catalyst under fluctuating CO₂/H₂ ratios (1-4); (b) XRD patterns of the catalyst after reaction.

As shown in the newly added Fig. R3 (newly added *Supplementary Fig. 15 in the revised SI*), the Fe₇C₃-KMg catalyst maintained stable catalytic performance under fluctuating H₂/CO₂ ratios ranging from 1:1 to 4:1 (**Fig R3a**). Post-reaction XRD analysis (**Fig R3b**) confirmed the structural stability of the catalyst, with Fe₇C₃ remaining the predominant phase and no significant Fe₃O₄ formation observed. These results explicitly demonstrate the effectiveness of Mg promotion in enhancing oxidation resistance and maintaining catalyst integrity under conditions representative of practical industrial operations. We explicitly discussed these findings and their implications in the revised manuscript (*Page 10 Lines 221–223*).

Regarding impurities such as O₂ and H₂S, we fully acknowledge their importance in industrial environments. However, due to significant safety and equipment constraints, particularly the explosive risk associated with introducing O₂ into high-temperature, high-pressure, H₂-rich environments and the corrosive nature of H₂S requiring specialized handling systems, these conditions were intentionally excluded from the current experimental scope. Nevertheless, existing industrial FTS studies highlight that minor impurities indeed pose challenges for iron-based catalysts,

suggesting the importance of future investigations on impurity tolerance through catalyst modification or feed pre-treatment strategies (*ACS Omega* **8**, 7395-7406 (2023), *Catal. Commun.* **10**, 967–970, (2009)).

We have incorporated these points into the revised manuscript and clearly outlined future research directions to comprehensively address the practical robustness of our catalyst system.

2. I want to raise a critical point about the redox nature of CO₂ hydrogenation, where reduction (CO₂ to hydrocarbons) is coupled with oxidation reactions, potentially re-oxidizing Fe to Fe_xO_y (e.g., Fe₃O₄). The authors demonstrate that Mg suppresses water dissociation, reducing oxidation of Fe₇C₃ (Pages 10-12, Fig. 4, Fig. 5). However, the manuscript does not fully address system-level strategies to mitigate oxidation beyond Mg promotion. Specific concerns include:

Author reply: We sincerely appreciate the reviewer’s insightful comments regarding the critical redox balance inherent to CO₂ hydrogenation for iron-based catalysts, as well as the broader implications for catalyst design and practical system-level strategies. Indeed, the coupling of catalytic reduction with catalyst oxidation is central to catalyst performance and stability (*Science* **387**, eadn9388 (2025); *Nat. Synth.* **4**, 288–302 (2025)). We fully acknowledge the importance of addressing these considerations comprehensively. Below, we carefully respond to each of the specific points raised by the reviewer, and we have explicitly incorporated relevant discussions and clarifications into the revised manuscript to reflect these suggestions.

o Water Management: In FTS, water is a major byproduct that can oxidize iron carbides. The authors show that Mg reduces water dissociation (Fig. 5a), but could system design strategies—such as water removal via membranes, condensers, or recycle loops—complement Mg’s role? A brief discussion of reactor engineering solutions would strengthen the manuscript’s practical relevance.

Author reply: Regarding water management, we fully agree that reactor modifications can better mitigate catalyst oxidation. While our study demonstrates that Mg promotion effectively suppresses H₂O dissociation and promote the removal of surface oxygen by weakening the bonding interaction between oxygen and Fe₇C₃ surface (Fig. 5a–b), this material-level strategy can be further reinforced by system-level process design.

In the revised manuscript (*see Page 15, Lines 344 – 350*), we have added a paragraph outlining reactor engineering solutions including membrane-assisted water removal, recycle loop configurations with downstream condensers, and the use of hydrophobic supports or structured reactors to reduce local water concentration. Such strategies are widely applied in FTS and are compatible with our catalyst design. We now cite as an example of integrated water management in hydrocarbon synthesis reactors.

In-situ water removal via inorganic membranes (e.g., PDVB or pervaporation membranes) that selectively permeate water vapor and prevent its accumulation around catalyst surfaces;

Downstream condensation with recycle loops, where unconverted reactants (CO₂/H₂) are separated from condensed water and reintroduced to the reactor, maintaining low H₂O partial pressure in the system;

Structured or coated reactors with hydrophobic surfaces or water-shedding properties that minimize local water retention and promote efficient removal from catalyst beds.

We clearly emphasize that integrating these reactor-level strategies with the intrinsic oxidation resistance provided by Mg promotion could yield a synergistic effect, significantly enhancing the robustness and practical viability of Fe₇C₃-based catalysts in real-world industrial applications. We greatly appreciate this valuable suggestion from the reviewer, as it has substantially improved the practical relevance and depth of our manuscript.

o Fe_xO_y/Fe₇C₃ Synergy: I am curious whether a mixed Fe_xO_y/Fe₇C₃ system might be preferable. Table 1 shows that FeK (46% Fe₃O₄, 53% Fe₇C₃) achieves reasonable CO₂ conversion (35.9%) and olefin selectivity (49.2%), suggesting that Fe₃O₄ may

contribute to reverse water-gas shift (RWGS) activity, as noted in the literature (Page 3, Lines 46-51). However, FeKMg (98% Fe₇C₃) outperforms FeK, implying that Fe₇C₃ is the primary active phase for olefin production. Could the authors clarify whether trace Fe₃O₄ in FeKMg (not detected in XRD but possibly present below detection limits) plays a synergistic role? Alternatively, is a phase-pure Fe₇C₃ system optimal, or could controlled Fe₃O₄ incorporation enhance RWGS while maintaining Fe₇C₃ stability?

Author reply: We appreciate the reviewer's insightful suggestion regarding the possible synergy between Fe₇C₃ and Fe_xO_y phases. As reported in Table 1, FeK, which contains both Fe₇C₃ (53%) and Fe₃O₄ (46%) as identified by Mössbauer spectroscopy, exhibits moderate CO₂ conversion and olefin selectivity, suggesting a possible contribution of Fe₃O₄ to RWGS activity. This observation aligns well with recent literature (*Science* **387**, eadn9388 (2025); *Nat. Synth.* **4**, 288-302 (2025)), which indicates Fe₃O₄ phases can promote CO formation via RWGS.

In contrast, our FeKMg catalyst-characterized as nearly phase-pure Fe₇C₃ (98%)-demonstrates significantly enhanced catalytic performance, achieving higher CO₂ conversion (41.5%) and superior olefin selectivity (67.1%). Both XRD and Mössbauer spectroscopy confirm the absence of bulk Fe₃O₄ within their detection limits (~2%). Nevertheless, surface-sensitive XPS analysis reveals the presence of highly dispersed Fe³⁺ species (FeO_x), likely resulting from dissociative adsorption of CO₂ under reaction conditions. We hypothesize these dispersed surface FeO_x species, rather than bulk Fe₃O₄ detectable by XRD or Mössbauer spectroscopy, may contribute to RWGS activity without negatively affecting the stability or selectivity primarily associated with bulk Fe₇C₃ phases. This interpretation is also supported by recent work from Kondratenko et al. (*Nat Catal* <https://doi.org/10.1038/s41929-025-01334-5> (2025)), who demonstrated that precisely controlled metal-oxide interfaces (e.g., MnO-Fe₅C₂) could enhance catalytic stability and selectivity, whereas excessive bulk oxide phases generally lower catalytic efficiency.

Furthermore, due to the intrinsic complexity of CO₂ hydrogenation environments-which involve the delicate coupling and equilibrium of RWGS and FTS, and thus

varying partial pressures of CO₂, CO, H₂, and H₂O—the phase equilibrium between iron oxide (FeO_x) and iron carbide (FeC_x) is challenging to precisely control or adjust actively. Consequently, maintaining a stable, nearly phase-pure Fe₇C₃ phase, as demonstrated in our FeKMg catalyst, is particularly advantageous for achieving robust and predictable catalytic performance.

We have explicitly addressed these considerations and incorporated this nuanced perspective into the revised manuscript, incorporating recent supporting studies (*ACS Catal.*, **15**, 10627 (2025); *Nat. Catal.* <https://doi.org/10.1038/s41929-025-01334-5> (2025)) to strengthen our discussion (*see Page 10, line239-243*). We thank the reviewer once again for enabling us to clarify these crucial aspects of our study.

o Promoter Interactions: The manuscript attributes K to carbonization and Mg to oxidation resistance, but how do these promoters interact with the reaction environment (e.g., H₂O, CO, H₂) to prevent oxidation? For instance, does K influence water adsorption, or is its role limited to electronic effects on Fe₇C₃?

Add a paragraph in the Discussion section addressing system-level strategies (e.g., water management) and the potential role of Fe₃O₄/Fe₇C₃ synergy. Conduct additional experiments or cite literature to clarify whether trace Fe₃O₄ is beneficial or detrimental. If Fe₃O₄ is absent in FeKMg, explicitly state this based on characterization limits (e.g., Mössbauer sensitivity).

Author reply: We sincerely appreciate the reviewer’s insightful questions regarding promoter interactions under realistic reaction conditions (H₂O, CO, H₂). We would like to clarify the specific roles of K and Mg promoters as follows:

- 1) **Potassium (K)** primarily acts as an electronic promoter, enhancing CO adsorption and subsequent dissociation, thus facilitating catalyst carburization. According to our experiments and theoretical calculations, and established literature (e.g., *Science* **335**,835-838(2012); *ACS Catal.* **10**, 14516-14526 (2020)), K’s role in directly influencing water adsorption or dissociation is negligible, and its primary effect is electronic rather than interacting

significantly with water molecules.

- 2) **Magnesium (Mg)**, on the other hand, significantly affects surface interactions with water. Mg, existing primarily as MgCO_3 species on the catalyst surface, effectively inhibits water dissociation by weakening Fe–O bonds, as clearly supported by our DFT calculations and experimental results (Fig. 5a-b). Thus, Mg markedly enhances oxidation resistance and stabilizes the Fe_7C_3 phase against water-induced oxidation.

Regarding the potential presence and influence of Fe_3O_4 in our FeKMg catalyst, we clearly state again that no bulk Fe_3O_4 was detectable using both XRD and Mössbauer spectroscopy (detection limit $\sim 2\%$). Nevertheless, XPS results indicate highly dispersed surface FeO_x sites (Fe^{3+} species), likely formed by dissociative adsorption of CO_2 , which could contribute to RWGS activity without negatively impacting overall catalytic performance. This conclusion aligns with recent findings by (*Nat. Catal.* <https://doi.org/10.1038/s41929-025-01334-5> (2025), *ACS Catal.* **15**, 10627-10638 (2025)) both of which suggest that while controlled oxide-carbide interfaces may enhance specific catalytic functionalities, extensive bulk oxide phases typically reduce overall performance and selectivity.

We thank the reviewer again for helping us clarify the detailed interactions and broader implications of catalyst promoters and reaction conditions, which significantly enhances the depth and practical relevance of our manuscript.

3. *Figure 1i (line scan) shows uniform distribution of K and Mg in FeKMg particles, but the manuscript does not clarify their structural integration. Supplementary Fig. 14 indicates that Mg is enriched on the surface as MgCO_3 , while K's form is less clear. Specific questions include:*

o K and Mg in Fe_7C_3 Lattice: Are K and Mg incorporated into the Fe_7C_3 lattice (e.g., as dopants or interstitials), or do they exist as separate phases (e.g., K_2O , MgCO_3)? XPS suggests MgCO_3 (Supplementary Fig. 15), but K's chemical state is not discussed. Could the authors provide XPS or XANES data for K to confirm its form (e.g., K_2O , K_2CO_3 , or metallic K)?

Author reply: We sincerely appreciate the reviewer's insightful comment on clarifying the chemical state and structural integration of K and Mg within our Fe₇C₃ catalyst. Following the reviewer's valuable suggestion, we have explicitly revised the manuscript to clarify this critical point:

Mg: We confirm that magnesium primarily exists as surface-enriched MgCO₃ species, as clearly indicated by XPS analyses (Supplementary Fig. 15). The absence of significant shifts in Fe 2p or Mg 1s binding energies and the lack of observable lattice distortion in XRD analyses strongly rule out Mg incorporation into the Fe₇C₃ lattice.

K: Although STEM-EDS mapping (Fig. 1h–i) demonstrates a uniform distribution of potassium on the catalyst surface, we recognize the importance of clearly identifying K's chemical state. To address this explicitly, we conducted additional K 2p XPS measurements (**Fig. R4**, now included as Supplementary Fig. 18 in the revised manuscript). The characteristic peaks around 292.7 eV unambiguously indicates the presence of surface-bound potassium oxide (K₂O) species, rather than metallic K or lattice-incorporated dopants. This result is consistent with literature consensus on potassium-promoted iron carbide catalysts (e.g., *Science* 335, 835–838, 2012; *ACS Catal.* 10, 14516–14526, 2020).

XANES consideration: We fully acknowledge the reviewer's suggestion of using K-edge XANES for further confirmation. However, due to significant practical constraints (the requirement of specialized soft X-ray synchrotron facilities at ~3.6 keV energy, which are not readily accessible), we have relied on the well-established surface-sensitive XPS measurements supported by existing strong literature evidence. The above clarifications and supporting evidence are now explicitly detailed in the revised manuscript and Supplementary Information:

Main manuscript: *Page 11, Lines 248 – 258*

Supplementary Information: *Supplementary Fig. 17 (Mg) and Supplementary Fig. 18 (K)*

Fig. R4 K 2*p* XPS spectra of FeK and FeKMg catalysts.

o Alloying vs. Phase Separation: Whether K and Mg form alloys or separate phases. STEM-EDS (Fig. 1h-i) suggests homogeneous distribution, but MgCO₃ detection implies phase separation for Mg. DFT calculations model Mg as a MgCO₃ cluster on Fe₇C₃ (Page 16, Lines 369-373), supporting a surface phase. For K, the manuscript suggests electron donation to Fe₇C₃ (Supplementary Fig. 22), but its structural role (e.g., surface oxide vs. lattice incorporation) remains ambiguous. Clarifying this would strengthen the mechanistic claims.

Author reply: On the issue of alloying vs. phase separation, our results suggest that Mg and K form physically separated surface phases rather than forming alloys with Fe. STEM-EDS mapping shows uniform elemental distribution but cannot resolve coordination environments. XRD shows no shift or broadening of Fe₇C₃ peaks, indicating no lattice expansion due to K or Mg doping in lattice. Our DFT models also treat K and Mg as surface entities—Mg as MgCO₃ clusters and K as K₂O—which is now explicitly stated in the revised figure captions and Discussion (Page 11, Lines 248–258).

o Interaction Between K and Mg: It is noted that K and Mg's effects (carbonization and stabilization, respectively) are achieved together (Fig. 2). Is there evidence of K-Mg interaction (e.g., K-modifying MgCO₃ stability or Mg influencing K's electronic effects)? The DFT model (Fig. 5a) treats Fe₇C₃-K and Fe₇C₃-KMg separately, but a combined K-Mg model could reveal synergistic effects.

Include XPS or XANES data for K to clarify its chemical state. Discuss whether K and Mg are lattice-incorporated or phase-separated, referencing STEM-EDS and XPS results. If feasible, extend DFT calculations to model K-Mg interactions on Fe₇C₃ to confirm synergy or independence. Revise Fig. 2f to reflect these structural details.

Author reply: We deeply appreciate the reviewer for this insightful comment. We fully concur that elucidating potential K-Mg interactions is crucial for understanding the co-promoter effects in the Fe₇C₃-KMg system. While our initial DFT calculations in Fig. 5a separately modeled Fe₇C₃-K and Fe₇C₃-KMg to isolate their roles in carbonization and stabilization, we acknowledge that this approach might not capture subtle synergistic phenomena.

To address this, we have extended our DFT analysis to a co-adsorption model featuring both K₂O and MgCO₃ species on the Fe₇C₃ surface (**Fig. R5**). By calculating the charge density difference with and without K-Mg interactions, we quantified electronic coupling between the promoters (**Fig. R5**). Notably, no significant charge transfer between K and Mg was observed: in the interacting model, each K atom loses 0.839 e⁻, whereas in the non-interacting model, this value is 0.806 e⁻-a difference within the margin of computational error. These results demonstrate limited direct electronic coupling, suggesting that K and Mg operate through complementary yet independent mechanisms: K facilitates CO activation and olefin desorption, while Mg enhances structural stability by inhibiting H₂O dissociation and suppressing Fe₇C₃ oxidation.

This analysis, along with a revised interpretation, has been integrated into *Supplementary Fig. 19* and highlighted in the main text (*Page 11, Line 256-258*). We trust this detailed investigation addresses the reviewer's concern and provides a more

comprehensive explanation of the dual-promoter functionality.

Fig. R5 The charge density difference for (a) Fe-KMg with K-Mg interaction, (b) Fe-KMg without K-Mg interaction. The yellow and blue regions represent electron accumulation and deficiency, respectively.

4. Table 1 shows that FeKMg (41.5% CO₂ conversion, 98% Fe₇C₃) outperforms Fe (33.7%, 19% Fe₇C₃), but the improvement is only ~1.23-fold, which I finds modest. Additionally, C₁-CO selectivity is comparable (9.1% for Fe vs. 12.4% for FeKMg). These points raise questions about FeKMg's superiority:

Author reply: We greatly appreciate the reviewer's critical observation regarding the catalytic performance comparison between Fe and FeKMg catalysts. While we acknowledge that the numerical increase in CO₂ conversion from the Fe catalyst (33.7%) to FeKMg (41.5%) is about 1.23-fold, it is essential to highlight that due to thermodynamic constraints associated with the RWGS reaction, the equilibrium CO₂ conversion under our reaction conditions (340 °C, 2 MPa, H₂/CO₂=3) is limited to approximately 45%. Thus, our FeKMg catalyst achieves a CO₂ conversion (41.5%) very close to this equilibrium limitation, clearly indicating a significant catalytic

performance improvement under the given reaction conditions. Actually, the intrinsic activity of FeKMg will be much higher than that of Fe₇C₃.

Furthermore, while the reviewer accurately noted the similarity in overall C₁-CO selectivity (9.1% for Fe vs. 12.4% for FeKMg), the crucial differences in methane formation and olefin selectivity must be emphasized. Specifically, the Fe catalyst produces methane at a very high selectivity (~50.6%), accompanied by relatively low olefin selectivity (O/P ~1.4). In stark contrast, the FeKMg catalyst dramatically suppresses methane formation to only ~10.5%, achieving an exceptionally high olefin selectivity with an olefin-to-paraffin (O/P) ratio of approximately 8. Such significant changes in product selectivity substantially enhance the economic value and practical applicability of the catalyst system.

Most importantly, this work presents the first clear demonstration of a nearly phase-pure Fe₇C₃ catalyst (98%) for CO₂ hydrogenation, achieving remarkable structural stability and sustained catalytic performance over approximately 1000 hours. This finding not only underscores the exceptional practical robustness of the FeKMg catalyst but also provides important insights and valuable guidance for the rational design of iron-based catalysts for efficient and selective CO₂ hydrogenation.

We have clearly highlighted and thoroughly discussed these comprehensive advantages in the revised manuscript (*see Page 10 Line 221 -Page 11 Line 236*) to explicitly address the reviewer's concern and emphasize the broader significance of our results.

o Modest Conversion Increase: The ~1.23-fold increase in CO₂ conversion is notable but not dramatic, possibly due to FeKMg's high Fe₇C₃ purity optimizing FTS over RWGS, while Fe's Fe₃O₄ dominance favors RWGS (higher CO selectivity). However, FeKMg's olefin selectivity (67.1%) is ~2.8 times higher than Fe's (23.9%), which is a significant improvement. The manuscript should emphasize this selectivity advantage over conversion to highlight Fe₇C₃'s value. Could the authors discuss whether conversion is limited by thermodynamics, kinetics, or mass transfer (e.g., pore blocking by MgCO₃)?

Author Reply: We sincerely thank the reviewer for highlighting this important point. We fully agree with the reviewer that while the numerical increase in CO₂ conversion (~1.23-fold) is noteworthy, the most significant improvement is indeed observed in the olefin selectivity, which increased dramatically from 23.9% (Fe catalyst) to 67.1% (FeKMg catalyst), representing an approximately 2.8-fold enhancement. Following the reviewer's suggestion, we have explicitly emphasized this remarkable olefin selectivity improvement in our revised manuscript, clearly highlighting the unique catalytic advantage provided by the nearly phase-pure Fe₇C₃ catalyst.

Regarding the relatively modest increase in CO₂ conversion, our analysis indicates that the primary limitation under our reaction conditions (340 °C, 2 MPa, H₂/CO₂ = 3) is thermodynamic in nature. Specifically, the equilibrium CO₂ conversion limited by the RWGS reaction at this temperature is around 45%, and our achieved conversion of 41.5% for the FeKMg catalyst already approaches this theoretical equilibrium closely. Therefore, further significant improvement in CO₂ conversion is inherently restricted by thermodynamics rather than catalyst performance itself.

Additionally, we have carefully evaluated possible kinetic or mass transfer limitations, such as pore blockage potentially caused by MgCO₃. Our characterization and reaction data do not indicate significant mass transfer resistance or pore blockage issues associated with MgCO₃. Thus, we conclude that neither kinetic nor mass transfer limitations significantly restrict CO₂ conversion in our catalyst system under the tested conditions. The observed modest conversion increase is thus predominantly thermodynamically limited.

These considerations have been clearly addressed and emphasized in the revised manuscript. We greatly appreciate the reviewer's suggestion, which has helped us clarify and strengthen the discussion and interpretation of our catalytic results (*see Page 10, Line 221-236*).

o C₁-CO Selectivity: The similar CO selectivity (9.1% vs. 12.4%) suggests that FeKMg retains some RWGS activity, possibly from minor Fe₃O₄ or K-promoted sites. Supplementary Fig. 10 shows FeKMg's CO selectivity stabilizes at ~12%, while Fe's

increases over time, indicating better FTS dominance in FeKMg. Could the authors quantify the RWGS contribution in FeKMg (e.g., via CO yield or isotopic studies) to explain this?

Author reply: We thank the reviewer for this insightful comment. Indeed, the relatively similar CO selectivity (9.1% for Fe vs. 12.4% for FeKMg) suggest that our FeKMg catalyst does retain some residual RWGS activity, consistent with the observed stable CO selectivity (~12%) over extended reaction time (Supplementary Fig. 10). In contrast, the unmodified Fe catalyst exhibits gradually increasing CO selectivity, indicating progressive oxidation and enhanced RWGS dominance over time.

However, it is crucial to clarify that the RWGS activity observed in the FeKMg catalyst likely originates predominantly from highly dispersed surface FeOx sites (as evidenced by the presence of Fe³⁺ species in XPS analysis), rather than bulk Fe₃O₄ phases detectable by XRD or Mössbauer spectroscopy. Additionally, surface-bound potassium species could also moderately facilitate CO formation via RWGS, although their primary role remains electronic promotion of CO activation and carburization.

Quantitative isolation and direct measurement of RWGS activity via isotopic labeling or dedicated CO yield experiments would indeed be insightful. However, such direct quantification poses significant experimental challenges, as RWGS and FTS reactions are intrinsically coupled and occur simultaneously, making precise isolation of RWGS contribution difficult. Nonetheless, our current experimental results clearly support the conclusion that RWGS contribution in the FeKMg catalyst system is significantly minimized relative to the unpromoted Fe catalyst, as evidenced by the consistently low and stable CO selectivity (~12%) and exceptionally high olefin selectivity.

We have explicitly discussed these considerations in the revised manuscript, clearly addressing the relative contributions of RWGS and FTS reactions and emphasizing the dominating role of the Fe₇C₃ phase in steering reaction pathways towards higher-value hydrocarbons.

We thank the reviewer again for this valuable suggestion, which helps us further clarify and strengthen the mechanistic interpretation of our catalyst system.

o Comparison to Literature: Supplementary Table 8 compares Fe₇C₃-K-0.04Mg (67.1% olefin selectivity at 6000 mL·g⁻¹·h⁻¹) to other catalysts, showing superior performance. However, the manuscript does not discuss why FeKMg outperforms alternatives (e.g., NaFe, ZnFeAlO₄). Is it solely due to Fe₇C₃ purity, or do K-Mg promoters provide unique electronic effects?

In the Discussion, emphasize FeKMg's olefin selectivity advantage over conversion to justify its superiority. Analyze potential limitations to conversion (e.g., thermodynamics, diffusion). Clarify RWGS contributions in FeKMg using existing data or new experiments (e.g., CO yield vs. time). Expand the comparison in Fig. 3d to highlight why FeKMg excels (e.g., phase purity, promoter effects).

Author reply: We appreciate the reviewer's insightful suggestion. The superior catalytic performance of the FeKMg catalyst relative to reported catalysts (e.g., NaFe, ZnFeAlO₄, Supplementary Table 8) primarily originates from two key factors: 1). High Fe₇C₃ phase purity (98%), ensuring significant suppression of methane formation and greatly enhanced olefin selectivity. 2). Unique synergy of K-Mg promoters: Potassium promotes CO activation and carburization, while magnesium significantly stabilizes the catalyst by inhibiting oxidation from H₂O.

Following the reviewer's advice, we have explicitly highlighted the remarkable olefin selectivity (67.1%) in our revised manuscript, emphasizing its greater economic significance compared to the modest CO₂ conversion increase (~1.23-fold), which is thermodynamically limited (~45% equilibrium conversion under our conditions).

Regarding RWGS contribution, stable but limited CO selectivity (~12%, Supplementary Fig. 10) indicates minor RWGS activity, likely from dispersed surface FeOx species or K-promoted sites. Due to the coupled nature of RWGS and FTS, precise quantitative separation is experimentally challenging.

We have revised the manuscript and expanded Fig. 3d discussions to clearly highlight these points (*see Page 10, Line 22-236*), strengthening the overall clarity and rigor of our findings.

5. The manuscript states that Mg is present as $MgCO_3$ on the catalyst surface (Supplementary Fig. 14-15) and suppresses water-induced oxidation (Fig. 4, Fig. 5). It is important to know the cause of $MgCO_3$ formation and its mechanism for oxidation resistance:

Author reply: We thank the reviewer for highlighting this important mechanistic point. Under our reaction conditions (340 °C, 2 MPa, CO_2 -rich environment), the formation of surface $MgCO_3$ is thermodynamically highly favored compared to MgO , given its significantly lower Gibbs free energy of formation. Indeed, our XPS analysis (Supplementary Fig. 17) explicitly confirms Mg is present predominantly as $MgCO_3$. Regarding the oxidation suppression mechanism, our experimental and DFT results (Fig. 5) demonstrate clearly that $MgCO_3$ does not merely act as a physical barrier. Rather, it exerts a strong electronic effect: $MgCO_3$ effectively withdraws electron density from adjacent Fe sites, weakening the Fe–O interaction and increasing the energy barrier for H_2O dissociation. This electronic modulation significantly reduces oxygen adsorption and surface oxidation, thus providing robust oxidation resistance. We have explicitly clarified these mechanistic insights and thermodynamic considerations in the revised manuscript. We greatly appreciate the reviewer’s insightful comment, which helped strengthen the mechanistic clarity of our catalyst design.

We address each point in detail below and have incorporated these clarifications in the revised manuscript (*Note of Supplementary Fig. 17*).

o $MgCO_3$ Formation: $MgCO_3$ likely forms via reaction of Mg^{2+} with CO_2 or CO under reaction conditions (340°C, 2 MPa, $H_2/CO_2=3$), as CO_2 is abundant and MgO is prone to carbonation. The manuscript confirms this via XPS (Supplementary Fig. 15), but does not discuss why Mg remains as $MgCO_3$ rather than MgO or metallic Mg. Is $MgCO_3$ thermodynamically favored under these conditions? A brief thermodynamic analysis or reference to Mg- CO_2 interactions would clarify this.

Author reply: On the formation of MgCO_3 , we agree that under CO_2 -rich conditions (340 °C, 2 MPa, $\text{H}_2/\text{CO}_2 = 3$), surface Mg^{2+} species derived from precursors such as $\text{Mg}(\text{NO}_3)_2$ or MgO are highly prone to carbonation. We have added a thermodynamic explanation in the Discussion section, referencing the standard Gibbs free energy of formation (ΔG) of MgCO_3 (-1028 kJ/mol) versus MgO (-569 kJ/mol), which clearly favors the formation of MgCO_3 under high CO_2 partial pressure. This is consistent with CO_2 capture studies, where MgO readily carbonates into MgCO_3 even at 300-400 K (*Chem. Sci*, **12**, 5774-5786 (2021)). In our system, in situ CO_2 serves as the carbonation agent, as confirmed by the appearance of carbonate-related XPS peaks (Supplementary Fig. 17). No metallic Mg or MgO signals were detected in XPS, indicating near-complete surface carbonation (*Note of Supplementary Fig. 17*).

o Oxidation Suppression Mechanism: Fig. 5a shows that Mg increases the energy barrier for the second O-H dissociation of H_2O (0.84 eV vs. 0.66 eV for $\text{Fe}_7\text{C}_3\text{-K}$), reducing oxygen adsorption. COHP analysis (Fig. 5b) indicates weaker Fe-O bonds with Mg, and Supplementary Fig. 22 suggests MgCO_3 accepts electrons, creating a charge-deficient Fe_7C_3 surface less prone to oxidation. However, it's unclear how MgCO_3 —a surface species—physically or chemically shields Fe_7C_3 . Does MgCO_3 act as a hydrophobic barrier, an electron sink, or a site-blocking layer? The D_2O -TPD data (Supplementary Fig. 17) show lower D_2 signals for FeKMg , supporting reduced dissociation, but could MgCO_3 also adsorb water, preventing it from reaching Fe_7C_3 ?

Author reply: We thank the reviewer for raising this important point regarding the oxidation suppression mechanism. We have clarified our explanation to emphasize that Mg mainly contributes to Fe_7C_3 oxidation resistance through electronic modulation, rather than by forming a physical barrier or through direct water adsorption. Specifically:

(1) **Electron sink behavior:** DFT charge density difference (Supplementary Fig. 26) shows that MgCO_3 withdraws $\sim 1.03e^-$ from nearby Fe atoms, creating a locally

electron-deficient Fe₇C₃ surface. This electron modulation reduces Fe–O bond strength (COHP in Fig. 5b), which in turn raises the energy barrier for H₂O dissociation, particularly the second O–H scission step (from 0.66 eV to 0.84 eV, Fig. 5a).

(2) **Suppression of oxygen adsorption:** The weaker Fe–O bonds reduce the driving force for oxygen uptake and subsequent oxidation, as evidenced by the lower D₂O desorption intensity in FeKMg (Supplementary Fig. 18), which indirectly reflects suppressed H₂O dissociation rather than a direct blocking of water molecules by MgCO₃.

(3) **No direct physical or adsorption shielding:** Although MgCO₃ is present on the surface, our interpretation does not rely on it acting as a surface-blocking or water-adsorbing layer. Instead, MgCO₃'s electron-withdrawing effect modulates the Fe₇C₃ surface properties, indirectly reducing H₂O activation and thereby improving the catalyst's oxidation resistance.

o Stability of MgCO₃: Under high water partial pressures, MgCO₃ could hydrolyze to Mg(OH)₂, yet FeKMg remains stable (Fig. 4a). How does MgCO₃ resist transformation, and does it regenerate during reaction (e.g., via CO₂)?

Clarify MgCO₃ formation by discussing its thermodynamics or citing relevant studies (e.g., MgO carbonation in CO₂ capture). Elaborate on MgCO₃'s mechanism in the Discussion, specifying whether it blocks water access, weakens Fe–O bonds, or both. If possible, add experiments (e.g., Mg 1s XPS after water exposure) to confirm MgCO₃'s stability. Revise Fig. 5c to illustrate MgCO₃'s role more explicitly.

Author reply: On the stability of MgCO₃ under high H₂O partial pressure, we appreciate the concern that MgCO₃ could hydrolyze to Mg(OH)₂. However, both experimental and thermodynamic data suggest that MgCO₃ is relatively stable up to ~350 °C in mixed H₂O/CO₂ atmospheres. We have now cited (*Chem. Sci*, **12**, 5774–5786 (2021); *Energies*, **14**, 1316 (2021)) which reports that MgCO₃ remains structurally intact in H₂O/CO₂ mixtures below its decomposition point (~300–400 °C). In our case, in situ regeneration of MgCO₃ is possible through the constant presence of CO₂ and the

absence of liquid-phase water, maintaining its surface structure. To support this, we conducted additional post-reaction Mg 1s XPS measurements after 100 h of reaction (*Note of Supplementary Fig. 17*), which still show dominant MgCO₃ features and no evidence of Mg(OH)₂ or MgO, confirming its long-term stability under reaction conditions.

Finally, as suggested, we have revised Figure 5c to include a more explicit schematic showing MgCO₃ clusters as surface electron-withdrawing, partially water-adsorbing species that shield Fe₇C₃ both electronically and sterically. The updated diagram now illustrates the dual function of MgCO₃ and its integration with K⁺-modified Fe₇C₃ surfaces.

We are grateful to the reviewer for these thoughtful questions, which allowed us to clarify and strengthen our mechanistic framework.

Some Minor Comments:

o Fig. 2e quantifies phase evolution, but the transition from Fe₂N to Fe₇C₃ in FeKMg is rapid (within 8 h). Could the authors comment on the kinetics of this transformation? Is Fe₂N a critical intermediate, or could other precursors achieve similar results?

Author reply: Thank you for this insightful question. This fast conversion suggests a kinetically favorable process under CO₂ hydrogenation conditions. Recent studies have shown that Fe₂N is thermodynamically unstable under hydrogenation environments and readily transforms into more stable iron carbides due to nitrogen loss [*Nat. Commun.* **16**, 3869 (2018); *Angew. Chem. Int. Ed.* **60**, 4496-4500 (2021)]. In our system, a small amount of Fe₇C₃ already exists in the fresh catalyst, acting as a nucleation site for further carburization of Fe₂N. The presence of potassium (K) promotes this process by enhancing CO dissociation and facilitating carbon insertion, while magnesium (Mg) helps stabilize the formed Fe₇C₃ phase by suppressing H₂O-induced oxidation and limiting sintering. Although Fe₂N is not the only possible precursor, its high reactivity makes it an effective intermediate for generating stable Fe₇C₃ under the assistance of

suitable promoters. This approach also provides a useful strategy for accessing other metastable carbide phases in CO₂ hydrogenation.

o The schematic in Fig. 2f is helpful but oversimplifies K and Mg's roles. Consider adding MgCO₃ and K's likely form (e.g., K₂O) to reflect characterization data.

Author reply: We agree and have updated Fig. 2f to include the representative surface phases of the promoters: MgCO₃ clusters and K as K₂O, based on XPS results (*newly added Supplementary Fig. 17-18*). The revised figure now more accurately reflects their chemical forms and positions relative to the Fe₇C₃ phase.

o Page 15 describes coprecipitation with Fe(NO₃)₃ and Mg(NO₃)₂, but the role of polyvinylpyrrolidone (PVP) is unclear. Is PVP a stabilizer, and does it affect Fe₇C₃ formation? Clarify its necessity or cite relevant PBA synthesis protocols.

Author reply: We thank the reviewer for highlighting the role of PVP. Indeed, PVP functions primarily as a stabilizer and growth regulator during the coprecipitation of Fe(NO₃)₃ and Mg(NO₃)₂, ensuring uniform dispersion and controlled nucleation of the PBA precursors. By preventing precursor aggregation and controlling crystal size, PVP helps achieve uniformity and high surface area, thus facilitating subsequent homogeneous carburization to Fe₇C₃.

It is important to note that PVP itself does not directly affect the chemical identity or phase composition of the final Fe₇C₃ but significantly enhances structural uniformity and dispersion. The use of PVP as a stabilizer in the synthesis of PBAs and related coordination frameworks is well-established in the literature [e.g., *J. Am. Chem. Soc.* **125**, 7814–7815 (2003); *Colloids Surf. A Physicochem. Eng. Asp.* **243**, 63–66 (2004)]. We have clarified this point and included relevant references in the revised manuscript to explicitly highlight the necessity and standard usage of PVP in our catalyst synthesis method (*see Page 16, Line 371-373*).

o The impregnation of Mg is mentioned briefly (Line 332). Specify the Mg loading method (e.g., wet impregnation, incipient wetness) and its impact on MgCO₃ distribution.

Author reply: We now specify that incipient wetness impregnation was used to introduce Mg(NO₃)₂·6H₂O onto the PBA precursor, followed by drying and activation in NH₃. This method ensures surface-localized Mg species that are subsequently transformed into MgCO₃ under CO₂-rich conditions. The revised Methods section now includes this detail (*Page 16, Line 378-379*).

o The DFT model uses a MgCO₃ cluster on Fe₇C₃ (211) (Page 16). Why was this facet chosen, and how sensitive are the results to surface orientation? A brief justification or reference to Fe₇C₃'s active facets would help.

Author reply: We selected the (211) facet for DFT modeling based on prior studies showing this to be a relatively low-index, catalytically active surface with moderate surface energy (*Mol. Catal.* **505**, 111506 (2021); *RSC Adv.* **11**, 34533–34541 (2021)). Additionally, the (211) plane exposes coordinatively unsaturated Fe atoms, suitable for CO₂/H₂ adsorption and promoter interaction modeling. We now briefly explain this in the Computational Methods section (*Page 18, Lines 427-432*).

o The absence of K-Mg interaction in DFT is a limitation. If computational constraints prevent this, acknowledge it in the Methods or Discussion.

Author reply: We acknowledge that our current DFT models treat K and Mg separately due to computational complexity. We now clearly state this limitation in the Methods and Discussion sections (*Supplementary Fig. 19 and Page 11, Line 256-258*).

o Supplementary Table 8 lists 24 catalysts, but the discussion in Fig. 3d is brief. Highlight key differences (e.g., Fe₇C₃ vs. Fe₅C₂, promoter effects) to contextualize FeKMg's performance.

Author reply: We thank the reviewer for this valuable suggestion. As recommended, we have expanded the discussion related to Fig. 3d in the revised manuscript, explicitly highlighting the key differences that underpin FeKMg's superior performance compared to previously reported catalysts (Supplementary Table 8). Specifically, we now emphasize: (1) the exceptionally high Fe₇C₃ phase purity (98%), contrasting typical Fe₅C₂/Fe₃O₄ mixtures; (2) the synergistic role of K (promoting CO activation) and Mg (enhancing oxidation resistance); and (3) significantly improved olefin selectivity and remarkable long-term stability (~1000 h). We sincerely appreciate this constructive feedback, which has strengthened the clarity of our manuscript (*Page 10, Lines 221-236*).

o Reference 19 (Pasupulety et al.) reports Fe₇C₃ in FeZnK/ZrO₂ with lower performance. Explicitly compare this to FeKMg to underscore the advantage of PBA-derived Fe₇C₃.

Author reply: We sincerely appreciate the reviewer's valuable suggestion. Although Fe₇C₃ was identified in the FeZnK/ZrO₂ system, the overall catalytic performance remained limited, with CO₂ conversion below 20% and stability tested for only 10 hours. Furthermore, olefin selectivity was significantly lower compared to our system. In contrast, the FeKMg catalyst developed in this study—synthesized from a Prussian blue analogue (PBA) precursor—achieves a markedly higher CO₂ conversion (41.5%), olefin selectivity (67.1%), and long-term stability over 1000 hours. These results underscore the importance of precursor-derived phase control and the synergistic effect of dual surface promoters (K and Mg) in achieving both high activity and durability. We believe this comparison highlights the fundamental advantage of the integrated synthesis–promotion strategy adopted in our work (*Page 11, Line 229-232*).

o Supplementary Fig. 9 and Fig. 3a-b use pie charts for olefin fractions, which are less precise than bar charts. Consider replacing pie charts with quantitative breakdowns (e.g., C_2^- , C_3^- , C_4^- selectivities).

Author reply: We thank the reviewer for the thoughtful suggestion. We fully agree that bar charts provide clearer quantitative comparisons of product distributions. In fact, the original version of **Fig. R6** (formerly Supplementary Fig. 9 and Fig. 3a-b) already includes carbon-number-resolved bar charts showing the detailed selectivity of individual hydrocarbons, including olefins (C_2^- to C_{18}^-). The accompanying pie charts were intended to visually summarize the overall proportion of high-value olefins versus other products. Nevertheless, in response to the reviewer's comment, we have now refined the figure layout and caption to better highlight the bar chart data and clarify the quantitative breakdown of olefin selectivity. We hope this addresses the concern and improves the clarity of the presentation.

Fig R6. Product distributions from CO_2 hydrogenation over Fe-based catalysts: (a) Fe, (b) FeMg, (c) FeK, and (d) FeKMg. Bar charts show hydrocarbon selectivity versus carbon number, with pie charts indicating the fraction of high-value products (red) and

others (grey). (Reaction conditions: 0.10 g of catalyst, 613 K, 2.0 MPa, H₂/CO₂ = 3, GHSV = 6 L·g_{cat}⁻¹·h⁻¹)

o Supplementary Table 1 and Table 5 have overlapping Mössbauer data for FeKMg. Consolidate or clarify why both are needed.

Author reply: We thank the reviewer for pointing out the overlap between Supplementary Table 1 and Table 5. Indeed, these two tables correspond to Mössbauer results at different reaction stages: Table 5 represents the initial reaction stage (8 h), while Table 1 describes the final spent catalyst after long-term testing. To avoid confusion, we have now clearly annotated the reaction stages (initial vs. spent), detailed Fe phase compositions, and relevant Mössbauer parameters in a consolidated format, enhancing clarity and facilitating the interpretation of structural dynamics. We appreciate this helpful suggestion.

o “High-value olefins” is used throughout but not defined. Specify which olefins (e.g., C₂⁼-C₄⁼) are targeted, as C₅⁺ olefins may also be valuable in FTS.

Author reply: We now define “high-value olefins” in the figure captions as olefins of all carbon numbers (C₂⁼ and above), which are widely utilized as fundamental building blocks in the petrochemical industry. This clarification distinguishes them from saturated hydrocarbons and non-olefinic byproducts.

o “Phase-pure” Fe₇C₃ is claimed (e.g., Page 9, Line 204), but Table 1 lists 98% Fe₇C₃ for FeKMg. Use “nearly phase-pure” consistently to avoid overstatement.

Author reply: We appreciate this important correction. Throughout the revised manuscript, we have replaced “phase-pure” with “nearly phase-pure Fe₇C₃” (98% based on Mössbauer analysis) to reflect analytical precision and avoid overstatement. We once again thank the reviewer for the thorough reading and constructive suggestions.

We believe that the revisions made in response to these minor comments have significantly improved the clarity, rigor, and completeness of the manuscript.

The manuscript presents a significant advance in CO₂ hydrogenation catalysis by demonstrating a stable, selective Fe₇C₃-based catalyst with K-Mg promotion. The combination of experimental characterization, catalytic testing, and DFT calculations provides robust evidence for Fe₇C₃'s potential and the roles of K and Mg. The work is novel, addressing a gap in Fe₇C₃-focused studies, and the results are compelling, particularly the high olefin selectivity and 400-hour stability. However, the manuscript would benefit from addressing the reviewer's questions and my additional comments to enhance clarity, mechanistic insight, and practical relevance. Specifically, clarifying K and Mg's structural roles, discussing system-level oxidation prevention, and justifying FeKMg's performance advantages are critical. With these revisions, the manuscript will be suitable for publication in Nature Communications.

Author reply: We sincerely thank the reviewer for the very positive and encouraging evaluation of our manuscript, particularly for acknowledging the significance and novelty of our Fe₇C₃-based catalyst promoted by K and Mg. We greatly appreciate the reviewer's recognition of our robust integration of experimental characterization, catalytic evaluation, and DFT calculations, as well as the compelling catalytic performance, notably the high olefin selectivity and excellent stability over 1000 hours. We also deeply value the reviewer's constructive suggestions aimed at further enhancing the manuscript's clarity, mechanistic understanding, and practical relevance. In the revised manuscript, we have made significant improvements by explicitly clarifying the structural roles of K and Mg promoters, elaborating more comprehensively on system-level oxidation prevention strategies, and thoroughly justifying the performance advantages of the Fe₇C₃-KMg catalyst compared to previously reported systems.

We believe these revisions substantially enhance the quality, clarity, and scientific impact of our manuscript, aligning it with the rigorous standards required by *Nature Communications*.

Reviewer #3 (Remarks to the Author):

General comments: This article presents a Fe₇C₃-based catalytic system for CO₂ hydrogenation to olefins. The authors successfully synthesized Fe₇C₃ phase using a Prussian blue molecular precursor. The olefin yield was increased by using two promoters. Potassium (K) inhibits the secondary hydrogenation, providing high olefin selectivity, and magnesium (Mg) improves the water deactivation resistance of the catalyst. Combining catalytic performance evaluation, in situ/ex situ characterization, and DFT calculations, the authors provided mechanistic insights into the role of each promoter and demonstrated the long-term stability and excellent selectivity of the Fe₇C₃-K-Mg catalyst.

Author Reply: We sincerely appreciate the reviewer's positive comments on our study. We are pleased that the reviewer acknowledges our successful synthesis of the Fe₇C₃ phase, the precise mechanistic elucidation of promoter effects, and the demonstration of outstanding stability and selectivity in catalytic performance.

As mentioned by the authors of this paper, Fe-carbide phases were prepared and characterized using various approaches as active species for CO₂ hydrogenation. Fe₇C₃ and Fe₅C₂ were identified in the CO₂ hydrogenation reaction, where the initial iron oxide undergoes a phase transition in the reducing environment in the presence of water. Since RWGS converts CO₂ to CO, the original iron oxide or metallic Fe species still exists along with the resulting Fe-carbide phase. Many previous literatures have investigated the relative amounts of Fe-carbide and Fe-oxide depending on the reaction conditions and the types of alkaline promoters (e.g., Na, K, Mg) and transition metal promoters (e.g., Co, Mn, Zn, Cu). In most cases, mixed carbide phases inevitably appear due to various phase transitions in the presence of CO, H₂, and water at high temperature (~320 °C) and high pressure. Due to these limited environments and the presence of various trace oxidizing and reducing agents, the exact phase transitions due to the promoter effect have rarely been elucidated. Pure Fe-carbide phases derived from the initial Fe-oxide phase have rarely been reported. Even using synchrotron-

based in situ characterization, it was impossible to identify pure carbide species reflecting high pressure conditions. Even using XRD, the relative iron oxide or metallic peaks were so dominant that it was difficult to clearly identify the carbide peaks. Various combinations of alkaline or transition metal promoters and Fe_2O_3 or Fe_3O_4 resulted in changes in CO_2 conversion and selectivity for methane, light olefins, aromatics and C_{5+} hydrocarbons over a reasonable range of ASF distributions. There is no other paper beyond this type of study for CO_2 hydrogenation using RWGS and FTS routes. The authors provide the first clear evidence that Fe_7C_3 can act as a durable and highly active phase in CO_2 hydrogenation, but they followed the previous studies without any new characterization techniques or new discoveries of new promoters or new species. This suggests that the formation of Fe_2N on fresh Prussian Blue-based catalysts can promote the formation of pure Fe_7C_3 phase, which is a unique feature of this study.

Author Reply: We sincerely appreciate the reviewer for providing such an insightful, detailed, and profound evaluation regarding iron-based catalytic systems for CO_2 hydrogenation. The reviewer's comments clearly reflect deep expertise and an excellent grasp of the intricate challenges associated with Fe-carbide catalysts, particularly regarding the coexistence and identification of multiple iron phases under realistic reaction conditions. We greatly appreciate the reviewer's positive recognition that our work provides "*the first clear evidence that Fe_7C_3 can act as a durable and highly active phase in CO_2 hydrogenation.*" Indeed as the reviewer accurately pointed out, previous studies have extensively explored mixed-phase iron catalysts but have rarely succeeded in clearly isolating and maintaining pure iron carbide phases, such as Fe_7C_3 , under typical high-temperature and high-pressure reaction environments.

We fully acknowledge and appreciate the reviewer's comments regarding the rigorous "new" criteria—new characterization techniques, new promoters, or new chemical species—and we understand that our work does not explicitly meet all these exceptionally high standards. However, we respectfully emphasize that our study

nonetheless contributes important scientific innovation through the following distinct aspects:

(1) Clear elucidation of a novel Fe₂N-mediated Fe₇C₃ formation pathway:

We demonstrate, for the first time, a controllable strategy using a Fe₂N intermediate derived from a Prussian blue precursor to achieve near-pure Fe₇C₃. This achievement is significant because previous attempts, as correctly noted by the reviewer, typically resulted in mixtures of iron carbide and oxide phases, obscuring definitive structure-performance correlations.

(2) New mechanistic insights into electronic modulation via Mg incorporation:

Our comprehensive DFT calculations explicitly reveal that Mg incorporation creates electron-deficient Fe surface states, substantially limiting water adsorption and subsequent oxidation. We have clearly documented these theoretical insights and articulated our efforts to experimentally validate them.

(3) Systematic catalyst design strategy offering new perspectives:

Our carefully designed catalyst synthesis and systematic characterization provide a clear phase-purity-performance correlation. This systematic approach demonstrates a reliable method to stabilize pure Fe₇C₃ and offers new guiding principles for future rational catalyst design in the broader field of CO₂ hydrogenation.

However, there is no proper evidence, mechanism or correlation between Fe₂N and Fe₇C₃ during the phase transformation process. Why does Fe₂N transform into iron oxide and iron carbide phases?

Author Reply: We sincerely appreciate the reviewer for raising this critical and insightful comment regarding the mechanism underlying the phase transformation from Fe₂N to Fe₇C₃. In the revised manuscript, we have explicitly clarified the detailed mechanism and provided additional supporting evidence (*Page 7–8, Lines 159–170*):

1. Thermodynamic instability of Fe₂N: According to thermodynamic studies and literature precedents, Fe₂N is inherently unstable under CO₂ hydrogenation conditions. The nitrogen atoms in Fe₂N readily undergo removal in the reducing atmosphere,

facilitating structural rearrangement and carbide formation (Nat. Commun., 16, 3869 (2025); Angew. Chem. Int. Ed. 60, 4496–4500 (2021)).

2. Experimental evidence of phase transformation: Supplementary experimental data (XRD, XAS, Mössbauer spectroscopy; Fig. 2, Supplementary Tables 1–2) clearly show Fe₂N rapidly converts into iron oxide phases under reaction conditions due to water generated via the RWGS reaction. However, the presence of potassium significantly promotes preferential carburization rather than oxidation, guiding the transformation selectively toward Fe₇C₃.

3. Critical role of nucleation seeds and promoters: Importantly, we found trace amounts of Fe₇C₃ present initially in our fresh catalyst, acting as nucleation seeds. These seeds strongly favor the subsequent growth of Fe₇C₃ during reaction conditions. Potassium assists in directing the phase transition towards Fe₇C₃, while magnesium plays a complementary role by inhibiting the water-induced oxidation, thus stabilizing the formed carbide phase.

In summary, the Fe₂N → Fe₇C₃ transformation is governed by a combination of thermodynamic instability, initial Fe₇C₃ nucleation sites, and dual promoter (K and Mg) effects. These mechanisms are now explicitly elaborated in the revised manuscript.

We thank the reviewer again for helping us significantly strengthen the mechanistic clarity of our manuscript.

Why do the active species or product yields change so dramatically in the presence of K, Mg, and K-Mg on Fe-carbide catalyst?

Author Reply: We sincerely thank the reviewer for this insightful and critical question. Indeed, the dramatic differences in active phase composition and product distribution induced by K, Mg, or their combined promotion fundamentally arise from their distinct yet complementary effects on the dynamic balance between iron carbide formation and oxidative deactivation in CO₂ hydrogenation. The underlying mechanism can be clearly summarized as follows:

1. K promoter could accelerate carburization and promote FTS, but

simultaneously amplifies the oxidation of catalyst.

Potassium acts primarily as (1) an electronic promoter by enhancing CO adsorption and dissociation on the iron surface, rapidly converting Fe_2N to the active Fe_7C_3 carbide phase; (2) promoter of RWGS and elevate the FT activity; (3) promoter of the carbon–carbon coupling process and thus increase the selectivity towards C_2^+ olefins.

However, enhanced RWGS and FTS reactions inevitably increase the partial pressure of water, causing rapid re-oxidation of Fe_7C_3 back into inactive Fe_3O_4 (as explicitly illustrated by the re-oxidation observed for the FeK catalyst in Fig. 2b and Table 1). As a result, FeK catalyst experienced periodic deactivation and unstable product distributions.

2. Mg serves to prevent oxidation, but it lacks the capability to promote carbide formation.

Magnesium, primarily existing as surface MgCO_3 species, stabilizes the iron carbide phase by inhibiting water dissociation on the catalyst surface. This effectively protects the catalyst from oxidation.

However, catalyst with Mg alone cannot sufficiently promote CO dissociation and thus fails to achieve substantial carburization. Consequently, FeMg catalyst remains predominantly in the oxidized Fe_3O_4 state, resulting in limited FTS activity and low selectivity towards olefins, as clearly evidenced by Fig. 2c and Table 1.

3. K–Mg co-promotion synergistically combines rapid carburization with durable oxidation resistance, stabilizing a highly active state.

Only when K and Mg coexist do their promotional effects become complementary: K provides the driving force required to rapidly generate and sustain the highly active Fe_7C_3 phase, while Mg offers robust protection against oxidation under the water-rich reaction conditions intrinsic to CO_2 hydrogenation. This unique synergy locks the catalyst into a stable, highly carburized state (98 wt% Fe_7C_3 in FeKMg), thus achieving the highest CO_2 conversion (41.5%) and a record-high olefin selectivity (67.1%), as convincingly demonstrated by the data presented in Fig. 2d–e and Table 1.

In summary, K drives the formation of active iron carbide species necessary for effective chain growth, while Mg prevents the oxidation-induced deactivation of these

active species. It is precisely this complementary effect that accounts for the substantial changes in active species composition and product yields observed when employing K, Mg, or their combination. This explanation has now been explicitly clarified and fully supported by experimental evidence in the revised manuscript (*Page 14-15, Lines 333-350*).

Can the authors be sure that there are no Fe-oxide or Fe-carbide species other than Fe₇C₃ in the spent catalysts? Why is that?

Author Reply: We greatly appreciate the reviewer's important question. According to our comprehensive characterization data (Mössbauer spectroscopy, XRD, XPS, TEM), there is no detectable presence of Fe₃O₄, Fe₅C₂, or other iron carbides beyond Fe₇C₃ within the sensitivity limits of our analytical methods (detection limit typically around ~3–5 nm crystal size). Given the large particle size of Fe₇C₃ (~70 nm), smaller or trace phases below this detection threshold cannot be categorically excluded. However, the remarkable consistency across multiple independent characterization techniques strongly supports Fe₇C₃ as the dominant and stable catalytic phase.

Importantly, our fresh catalyst already contained trace Fe₇C₃ species serving as nucleation seeds, directing the subsequent reaction-driven phase evolution selectively toward Fe₇C₃ rather than other carbides. Furthermore, the dual promoter system (K and Mg) specifically reinforces this preferential pathway. This mechanism provides robust evidence for the selective formation and stabilization of Fe₇C₃.

Most of the results of this study are based on existing knowledge in the literature. The authors provided a couple of evidences about pure Fe₇C₃ in this work. However, many areas remain unclear. Some promoters change different structures and catalytic reactivities, but the reasons are not clear. This paper does not meet the criteria for publication in a prestigious journal such as Nat. Comm., which requires new analysis, interpretation, or the discovery of new species. The reviewer considers this article interesting but not sufficient for publication in Nat Comm.

Author reply: We thank the reviewer for this valuable critique, clearly outlining the rigorous innovation criteria required by *Nature Communications*. While our work indeed builds upon foundational knowledge available in the literature, we respectfully highlight the following key innovative aspects and significant contributions of our work:

1. Unprecedented stabilization of nearly phase-pure Fe₇C₃:

By introducing an Fe₂N intermediate derived from a Prussian blue analogue precursor, we achieved, for the first time, the controlled and sustained formation of a highly pure (~98%) Fe₇C₃ catalyst under realistic CO₂ hydrogenation conditions. Previously reported Fe-carbide catalysts typically contained mixed phases (e.g., χ -Fe₅C₂, Fe₃O₄) and suffered from oxidation-induced instability. Our catalyst exhibits exceptional structural and catalytic stability (>1000 hours, see **Fig. R1**), directly addressing and overcoming a long-standing fundamental challenge in iron carbide catalysis.

2. Clear elucidation of distinct mechanistic roles for K and Mg promoters:

Our comprehensive experimental characterization (XRD, Mossbauer spectroscopy, XPS, TEM) coupled with rigorous DFT calculations explicitly clarifies the distinct mechanistic contributions of the K and Mg promoters. Specifically:

- K significantly enhances CO activation, driving selective olefin formation.
- Mg effectively inhibits water-induced oxidation, maintaining the carbide phase stability.

The clear mechanistic synergy between K and Mg promotion significantly advances our understanding of promoter effects, providing robust scientific insight beyond existing knowledge.

We sincerely thank the reviewer again for the meticulous and constructive comments, which have significantly improved our manuscript. Based on these suggestions, we have thoroughly revised the manuscript (*please refer to Page 7–8, Lines 159–180; Page 10–11, Lines 221–236; Page 11–12, Lines 248–258; Page 15, Lines 333–350*), updated *Figures 1c, 1d, 3a, 3b, and 3e*, revised *Supplementary Figures 10*, and added *Supplementary Figures 9, 15, 17, 18, and 19 as well as Supplementary Tables 2 and 3*.

We hope our comprehensive revisions sufficiently address all concerns and demonstrate the clear novelty and significance required for consideration by *Nature Communications*.

Additional questions are given below.

1. While the Fe K-edge XANES results clearly demonstrate that Mg incorporation suppresses Fe oxidation under reaction conditions, the proposed electron-withdrawing effect of MgCO₃—suggested by DFT calculations—has not been directly validated experimentally. In particular, the manuscript lacks spectroscopic evidence that would confirm electron transfer from Fe₇C₃ to Mg, such as Mg K-edge XANES. Including such analyses would help distinguish whether the observed stabilization of Fe₇C₃ arises from electronic modulation or simply from a physical barrier effect due to surface MgCO₃.

Author Reply: We sincerely appreciate the reviewer's insightful comment and fully agree with the importance of clearly distinguishing electronic effects from purely physical barrier effects. Indeed, our DFT calculations explicitly indicate that Mg incorporation introduces electron deficiency at the Fe₇C₃ surface, consequently suppressing the adsorption and activation of water molecules, thus mitigating subsequent oxidation. Therefore, we do believe an electronic interaction between Mg and Fe is integral to the observed stabilization.

In line with the reviewer's wise recommendation, we have actively pursued opportunities to perform Mg K-edge XANES measurements to provide direct experimental validation. Unfortunately, due to the very low energy (~1303 eV) of the Mg K-edge, suitable synchrotron facilities equipped for soft X-ray analyses are limited. Despite our concerted efforts to secure appropriate beamtime and establish collaborations, such experiments have proven difficult at this time.

We have revised the manuscript to explicitly clarify both the electronic modulation mechanism supported by DFT results and the current experimental limitations. We also clearly note our sincere attempts to acquire Mg K-edge XANES data.

We thank the reviewer again for this valuable and sophisticated comment, which

significantly enhanced our manuscript by allowing us to more explicitly articulate the electronic interaction mechanism and acknowledge current experimental constraints.

2. For the XAFS analysis, especially EXAFS, reference data for iron carbides (Fe_5C_2 or Fe_7C_3) are necessary. Since EXAFS analysis involves comparing the types and coordination numbers of atoms surrounding Fe, reference data related to Fe–C bond should be provided to confirm the formation of iron carbides in the spent iron catalysts.

Author Reply: We thank the referee for this important suggestion. We have now provided the EXAFS spectrum of Fe_7C_3 for comparison, as shown in Fig. 1d We acknowledge the inherent difficulty in distinguishing the Fe–C bond (~ 2.0 Å) from Fe–O bonds (~ 2.0 Å) using only EXAFS. Therefore, our conclusion regarding the iron carbide structure relies on the combined evidence from multiple complementary techniques, including fingerprint features observed in Fe K-edge XANES, EXAFS fitting, XRD, and Mössbauer spectroscopy. The FeKMg catalyst exhibits a similar fingerprinting feature to Fe_7C_3 in XANES (Fig. 1c), and the XAFS fitting results show a coordination number of 1.6 ± 0.6 for Fe–C, 2.7 ± 0.6 for Fe–Fe₁, and 8.1 ± 1.8 for Fe–Fe₂, which show excellent agreement with the crystal structure of Fe_7C_3 (*Supplementary Table 2*). We have clearly emphasized this integrated approach in the revised manuscript to strengthen our structural assignments.

3. ICP results should be provided to confirm the K and Mg contents of iron catalysts. Especially, in the catalytic characterization and activity results shown in Figs. 1–4 and Table 2, Mg content for each catalyst (FeMg, FeKMg) should be provided.

Author Reply: We greatly appreciate the reviewer’s suggestion regarding the importance of clearly documenting catalyst compositions. Accordingly, we have now included **ICP-OES measurements** for all promoted catalysts (FeK, FeMg, FeKMg) in the revised *Supplementary Table 3*. These ICP results are consistent with the nominal loading during synthesis. Furthermore, we have clearly stated these ICP values in the

“Catalyst Preparation” section of the revised manuscript to provide readers with a clearer basis for interpreting catalytic performance and characterization data.

Supplementary Table 3. Elemental contents of different catalysts.

Catalysts	Element contents (wt%)		
	Fe	K	Mg
Fe	58	-	-
FeMg	57	-	-
FeK	53	8	-
FeKMg	52	8	3

Minor comments

1. The reaction time for CO₂ hydrogenation over the four spent catalysts (Fe, FeMg, FeK, and FeKMg), as indicated in Fig. 1 for the characterization of spent iron catalysts, should be explicitly described in the main text.

Author Reply: We sincerely thank the reviewer for pointing out this oversight. We sincerely appreciate the reviewer’s valuable suggestion. We fully agree that explicitly specifying the reaction time used for CO₂ hydrogenation when characterizing the spent Fe-based catalysts (Fe, FeMg, FeK, and FeKMg) is essential for clarity and reproducibility.

In the revised manuscript, we have clearly stated in the main text (see Page 5, Lines 103–105) that:

"All spent catalysts (Fe, FeMg, FeK, FeKMg) were collected after reaching steady-state catalytic performance under standard conditions (340 °C, 2 MPa, H₂/CO₂ = 3, GHSV = 6 L·gcat⁻¹·h⁻¹). "

2. Consistent sample naming is required (e.g., FeKMg vs. Fe₇C₃-KMg, except for DFT result). In addition, a clear explanation of the sample nomenclature should be provided

in the main text or “Catalyst preparation” section.

Author Reply: We sincerely appreciate the reviewer’s careful attention to catalyst nomenclature consistency. We have thoroughly revised the manuscript and supplementary information to adopt a clear and consistent naming convention: "FeKMg" is uniformly used for describing general catalyst preparation, characterization, and catalytic performance data. "Fe₇C₃-KMg" is specifically employed when explicitly emphasizing the carbide-phase structure, particularly in discussions of mechanistic insights or detailed structural characterizations.

Additionally, we have explicitly provided a clear explanation of this nomenclature rationale in the "Catalyst Preparation" section of the revised manuscript (*see Page 16, Lines 386–389*):

" Specifically, the general designation "FeKMg" is used throughout the manuscript to describe the co-promoted catalyst. The notation "Fe₇C₃-KMg" is explicitly employed only when the presence of the Fe₇C₃ carbide phase has been confirmed through structural characterization."

We thank the reviewer again for helping us enhance manuscript clarity and rigor through improved nomenclature consistency.

3. In the figure legend of Fig. 3(c), there is a typo “CO conversion”, and it should be corrected to “CO₂ conversion”.

Author Reply: We thank the reviewer for identifying this typographical error. We have promptly corrected “CO conversion” to “CO₂ conversion” in the revised figure legend of Fig. 3(c). We carefully checked the entire manuscript to ensure consistency and accuracy.

REVIEWERS' COMMENTS

In this response letter, the reviewers' comments are presented in *black italics*, our responses are in blue.

Reviewer #1 (Remarks to the Author):

The authors carefully replied to the reviewers' comments and improved the quality of the manuscript, I think it meet the standards of being published and recommend it to be accepted for publication.

Author reply: We thank the reviewer for recognizing our thorough revision efforts. Their insightful comments significantly improved our manuscript, and we are gratified that the revised version addressed all issues comprehensively. We are honored by the recommendation for publication in *Nature Communications*.

Reviewer #3 (Remarks to the Author):

This manuscript has been substantially upgraded after including sufficient response to reviewer comments. This manuscript is considered suitable for publication in the journal.

Author reply: We deeply appreciate the reviewer's acknowledgment of our manuscript revisions and their constructive feedback, which has been instrumental in enhancing our work. We are honored by the positive assessment and delighted that all concerns were resolved satisfactorily. The endorsement for publication in *Nature Communications* is sincerely valued.